# Do Neural Operators Forget Geometry?
## The Forgetting Hypothesis in Deep Operator Learning

**Yanming Xia** [1]    **Angelica I. Aviles-Rivero** [2]

## Abstract

Neural operators perform well on structured domains, yet their behaviour on irregular geometries remains poorly understood. We show that this limitation is not merely an encoding issue, but a depth-wise failure mode inherent to deep operator architectures. We formalise the *Geometric Forgetting Hypothesis*: due to the Markovian structure of operator layers and their reliance on global mixing mechanisms, neural operators progressively lose access to domain geometry as depth increases. Using layer-wise geometric probing, we demonstrate that both spectral and attention-based operators systematically lose geometric fidelity. We show that this geometric forgetting degrades accuracy, stability, and generalisation. To counteract it, we introduce a lightweight *geometry memory injection* mechanism that restores geometric constraints at intermediate depths with minimal architectural overhead. This simple intervention consistently mitigates forgetting and exposes a *geometric shortcut* instability in transformer-based operators, revealing that geometric retention is a structural requirement rather than a design choice.

## 1. Introduction

Partial Differential Equations (PDEs) are fundamental to modeling complex physical systems, from aerodynamics to thermodynamics (Roubicek, 2005). While traditional numerical solvers offer high accuracy, their computational cost is often prohibitive (Umetani & Bickel, 2018). Neural Operators (Lu et al., 2021; Wen et al., 2022; Tran et al., 2023; Kovachki et al., 2023; Li et al., 2024; Kossaifi et al., 2025)

[1]Qiuzhen College, Tsinghua University, Beijing, China [2]Yau Mathematical Sciences Center, Tsinghua University, Beijing, China. Correspondence to: Yanming Xia <xiaym23@mails.tsinghua.edu.cn>, Angelica I. Aviles-Rivero <aviles-rivero@tsinghua.edu.cn.>.

*Proceedings of the $43^{rd}$ International Conference on Machine Learning*, Seoul, South Korea. PMLR 306, 2026. Copyright 2026 by the author(s).

have emerged as a paradigm-shifting alternative, learning mappings between infinite-dimensional function spaces to enable rapid, zero-shot evaluation.

Despite this progress, a major challenge remains: robust generalisation to irregular geometries. Real-world domains are rarely structured, yet most neural operator architectures implicitly assume regular grids. Recent works address this through increasingly sophisticated geometry handling strategies, including coordinate transformations (Li et al., 2023a; Taylor et al., 2025; Li et al., 2025), graph-based operators on unstructured meshes (Li et al., 2023b; Bryutkin et al., 2024; Li & Zhe, 2025), and Transformer-based models that encode geometry into tokens or continuous fields (Wu et al., 2024; Wang et al., 2024). A common element across these approaches is the repeated *conditioning* of hidden layers on geometric information.

In this work, we provide a principled explanation for why such geometry conditioning is necessary. We identify a structural phenomenon in neural operators, which we term the **Geometric Forgetting Hypothesis**. We show that the global mixing mechanisms underlying spectral and attention-based operators (FFT in FNO (Li et al., 2020), self-attention (Vaswani et al., 2017)) induce a progressive loss of geometric information as representations propagate through depth. This behaviour follows directly from the Data Processing Inequality applied to the Markovian structure of operator layers, causing deep representations to approach statistical independence from the domain geometry unless geometry is explicitly reintroduced.

We validate this hypothesis through layer-wise geometric probing and spectral analysis, revealing systematic geometric information decay in both spectral and attention-based backbones. Our analysis further uncovers a instability in Transformer-based operators, which we term the **Geometric Shortcut** (Geirhos et al., 2020; Liu et al., 2022): under late geometry injection, the optimiser bypasses the operator backbone, leading to feature collapse. This explains the sensitivity of Transformer operators to injection depth and shows that early geometric integration is a structural requirement rather than an architectural choice.

**Contributions.** This work provides *the first principled analysis* of how geometry propagates through deep neural operators and reveals a previously unnoticed structural limitation of standard architectures. Notably, we highlight:

● We identify and formalise a new phenomenon, termed **Geometric Forgetting**: an information-theoretic consequence of the Markovian structure of neural operator layers, whereby global mixing operations progressively erase geometric information across depth.

● We introduce practical **diagnostic probes** that quantify this layer-wise geometric information decay and empirically validate the phenomenon across spectral and attention-based operators, revealing a instability in Transformer-based models that we term the **Geometric Shortcut**.

● We propose a minimal, architecture-agnostic **Geometry Memory Injection** mechanism that restores geometric information flow, and through a control study with the Laplace Neural Operator, distinguish between **intrinsic** and **extrinsic** geometric memory.

## 2. Related Work

**Neural Operators.** Neural Operators have emerged as a powerful paradigm for learning solution operators of PDEs (Wen et al., 2022; Tran et al., 2023; Li et al., 2024; 2023a; Wu et al., 2024; Wang et al., 2024). The Fourier Neural Operator (FNO) (Li et al., 2020) pioneered this direction by parameterising the integral kernel in the Fourier spectral domain, enabling efficient global convolution with quasi-linear complexity. Numerous variants have since extended this framework (George et al., 2024; Guibas et al., 2022; Tran et al., 2023). A complementary direction is the Laplace Neural Operator (LNO) (Cao et al., 2024), which replaces the Fourier transform with a Laplace transform and explicitly encodes initial conditions through a pole–residue formulation, effectively incorporating boundary information into its parametrization. More recently, Transformer-based neural operators such as Transolver (Wu et al., 2024), Transolver++ (Luo et al., 2025), and CViT (Wang et al., 2024) lift inputs into geometry-aware tokens or continuous neural fields and employ global self-attention to model long-range physical interactions. Despite these architectural differences, a common trait of spectral and attention-based operators is the reliance on global mixing mechanisms that do not explicitly constrain how geometric information propagates through depth. Existing works primarily focus on how geometry is *encoded at the input*, leaving unexplored how geometry is *retained* across successive operator layers.

**Handling Irregular Geometries.** Adapting neural operators to complex, non-rectangular domains remains a primary challenge. Current approaches largely focus on the encoding stage (Sitzmann et al., 2020). One line of work employs coordinate transformations, such as Geo-FNO (Li et al., 2023a), diffeomorphism-based methods (Zhao et al., 2025) or optimal transport (Li et al., 2025), to map irregular physical domains into regular latent grids where standard FFTs can be applied. Another approach integrates graph-based methods, as seen in GNO (Anandkumar et al., 2020), GINO (Li et al., 2023b) or use graph neural operator (Sarkar & Chakraborty, 2025b;a), to handle unstructured meshes directly before projecting to a latent space. While some of these advanced architectures implicitly integrate geometric information into deeper layers, these design choices remain largely heuristic. Existing literature treats this deep injection merely as a means of increasing model capacity or context. The fundamental question of *why* continuous geometric conditioning is necessary remains unexamined.

**Memory Effects and Forgetting.** Memory effects in physics-informed modelling traditionally refer to temporal history dependence. This has been rigorously studied through the Mori–Zwanzig formalism for modelling unresolved scales in dynamical systems (Zwanzig, 1961; Mori, 1965; Chorin & Hald, 2000; Stinis, 2012). More recently, long-range temporal and spatial dependencies in sequence modelling have been addressed through State Space Models (SSMs) that explicitly encode memory into the architecture (Gu et al., 2022; Gu & Dao, 2023; Buitrago Ruiz et al., 2025). A closely related phenomenon appears in deep learning theory. It has been shown that repeated message passing in recurrent and graph architectures leads to *information collapse* and *over-smoothing*, where high-frequency components of the signal are progressively attenuated and hidden representations converge to low-dimensional subspaces (Bengio et al., 1994; Oono & Suzuki, 2020).

**Our Work: Beyond geometry encoding.** In contrast to prior work that focuses on how geometry is *encoded*, we investigate whether neural operators *retain* geometric information once it enters the network. We show that, due to their Markovian structure and reliance on global mixing layers, standard neural operators are structurally predisposed to progressively lose access to geometry with depth. From this perspective, geometry conditioning is not a heuristic design choice, but a necessary mechanism to counteract an inherent information dissipation within the architecture.

## 3. Methodology

We now formalise the learning setting and introduce the structural phenomenon that motivates our approach, namely the progressive loss of geometric information across depth in standard neural operators.

*Figure 1.* **Geometric forgetting as an architectural information-flow phenomenon.** Left: In standard neural operators, geometry $G$ is only provided at the input. The hidden states $\{V_l\}_{l=1}^{L}$ form a Markov chain $G \to V_1 \to \cdots \to V_L$, implying that geometric information cannot increase with depth. In global mixing layers (FFT, self-attention), this information decays severely, leading to *geometric forgetting*. Right: Geometry memory injection breaks this Markov structure by explicitly reintroducing $G$ into intermediate layers through $V_{l+1} = \mathcal{M}_l(\mathcal{O}_l(V_l), G)$, preserving geometric information across depth.

### 3.1. Problem Statement

Let $(\mathcal{D}, \mathcal{A}, \mathcal{U})$ be a probability space. A *physical domain* is a random compact subset $D \subset \mathbb{R}^d$ with Lipschitz boundary $\partial D$, drawn from a distribution $\mathcal{D}$. We consider a parametric PDE defined on a realisation $D$:

$$\begin{cases} \mathcal{L}_a(u)(x) = 0, & x \in D, \\ \mathcal{B}(u)(x) = 0, & x \in \partial D, \end{cases} \qquad (1)$$

where $a \in \mathcal{A} \subset L^2(\mathbb{R}^d; \mathbb{R}^{d_a})$ is a parameter field, $\mathcal{L}_a$ is a differential operator, and $\mathcal{B}$ is a boundary operator. We assume (1) admits a unique solution $u \in \mathcal{U} \subset L^2(D; \mathbb{R}^{d_u})$ for each $(a, D)$. The solution operator is $\Psi : \mathcal{A} \times \mathfrak{D} \to \mathcal{U}$, where $\mathfrak{D}$ is the space of admissible domains.

We seek to learn an approximation $\Psi_\theta \approx \Psi$ using a neural operator, with a focus on *geometric generalisation*: accurate prediction on domains $D_{\text{test}} \sim \mathcal{D}'$ distinct from those seen during training $D_{\text{train}} \sim \mathcal{D}$.

### 3.2. Neural Operator as a Markov Chain and Geometric Forgetting

Let $\mathcal{X} \subset \mathbb{R}^d$ be a fixed, bounded reference set (e.g. a bounding box containing all domains). For a domain $D$, we define a *geometry encoding* as a measurable function

$$G_D : \mathcal{X} \to \mathbb{R}^{n_g}, \qquad G_D \in \mathcal{G}, \qquad (2)$$

where $\mathcal{G}$ is a Banach space of encoding functions.

Typical representatives of $\mathcal{G}$ include indicator masks $G_D^{\text{mask}}(x) = \mathbf{1}_D(x)$, signed distance fields $G_D^{\text{sdf}}(x) = (1 - 2 \cdot \mathbf{1}_D(x)) \cdot \text{dist}(x, \partial D)$, coordinate embeddings $G_D^{\text{coord}}(x) = x$, or their concatenation $G_D = (G_D^{\text{mask}}, G_D^{\text{sdf}}, G_D^{\text{coord}})$, which is the encoding used in our experiments.

The random variable $G := G_D$, where $D \sim \mathcal{D}$, represents the geometric information available to the model. A neural operator consists of $L$ layers. Let $V_0 = \eta(a, G)$ be an initial lifting. For $l = 0, \ldots, L - 1$,

$$V_{l+1} = \mathcal{O}_l(V_l), \qquad \mathcal{O}_l : \mathcal{H}_l \to \mathcal{H}_{l+1}. \qquad (3)$$

Since each layer $\mathcal{O}_l$ does not depend explicitly on $G$, for every $l$ we have the Markov chain $G \to V_l \to V_{l+1}$.

**Proposition 3.1** (Information non-increase through depth). *Under the update rule* (3),

$$I(G; V_{l+1}) \le I(G; V_l), \qquad l = 0, \ldots, L - 1,$$

*where $I(\cdot; \cdot)$ denotes mutual information.*

*Proof.* Because $V_{l+1}$ is a measurable function of $V_l$ alone, it is conditionally independent of $G$ given $V_l$. Hence $G \to V_l \to V_{l+1}$ is a Markov chain, and the result follows from the Data Processing Inequality. $\square$

**Corollary 3.2** (Cumulative information decay). *By induction using Proposition 3.1,*

$$I(G; V_L) \le I(G; V_{L-1}) \le \cdots \le I(G; V_0).$$

Corollary 3.2 reveals a structural limitation of standard neural operators: unless geometry is explicitly reintroduced at intermediate layers, the architecture itself enforces progressive loss of geometric information. In particular, deep stacks of global mixing layers inevitably drive the representation towards statistical independence from the domain geometry. We refer to this phenomenon as *geometric forgetting*.

### 3.3. Quantifying Forgetting: Diagnostic Measures

Since direct computation of $I(G; V_l)$ is intractable, we introduce two measurable proxies. They detect two different kinds of geometric forgetting.

**Definition 3.3** (Layer-wise geometric fidelity). Let $D_l : \mathcal{H}_l \to L^2(\mathcal{X})$ be a trained auxiliary decoder estimating $\mathbb{E}[\mathbf{1}_D \mid V_l]$, i.e. reconstructing indicator mask $G_D^{\text{mask}}$ from latent feature $V_l$. Define

$$\epsilon_l := \mathbb{E}\|D_l(V_l) - G_D^{\text{mask}}\|_{L^2}^2.$$

An increasing sequence $\{\epsilon_l\}$ provides empirical evidence consistent with geometric information loss. We termed this as *macroscopic* geometric information loss.

**Definition 3.4** (Spectral power distribution of geometric features). Let $\mathcal{F}$ denote the Fourier transform and define

$$\rho_l(\kappa) := \frac{\sum_{\kappa \leq |k| \leq \kappa+1} |\mathcal{F}[v_l](k)|}{\sum_k |\mathcal{F}[v_l](k)|}.$$

We interpret the spectral power distribution as a proxy for *boundary* information loss. By the properties of the Fourier transform, spatial features of characteristic length scale $\lambda$ correspond to spectral energy concentrated around wavenumber $k = S/(2\lambda)$ where $S$ is resolution. Therefore, frequency $k = S/2$ captures the 'vertical' step of the discretized binary mask while frequency $k = S/4, S/8$ correspond to more smoothed staircase approximations like $G_D^{\text{sdf}}$ with wavelengths of 4 and 8 pixels.

**LNO can model boundary effect.** A periodic signal $u(t) = \sum_{l=-\infty}^{\infty} \alpha_l e^{i\omega_l t}, t \sim t + T$ processed after Laplace convolution yields (Cao et al., 2024):

$$u_1(t) = \sum_{n=1}^{N} \gamma_n e^{\mu_n t} + \sum_{l=-\infty}^{\infty} \lambda_l e^{i\omega_l t} \tag{4}$$

Thus, Laplace convolution can represent transient dynamics, which translate to boundary effect in geometry related problems. We conjecture that this effective modeling of boundary effect renders LNO *implicit memory* that also help the retention of geometry information. In contrast, Fourier convolution can only represent periodic signals, effectively 'washing out' boundary effect even macroscopic geometry information is preserved.

### 3.4. Geometry Memory Injection: A Compensatory Mechanism

To counteract geometric forgetting, we introduce an external *geometry memory pathway* that allows each layer to depend explicitly on $G$.

We modify the layer update to

$$V_{l+1} = \mathcal{M}_l\big(\mathcal{O}_l(V_l), G\big), \tag{5}$$

where $\mathcal{M}_l : \mathcal{H}_{l+1} \times \mathcal{G} \to \mathcal{H}_{l+1}$ is a memory integration operator.

This update violates the Markov condition $G \to V_l \to V_{l+1}$, allowing geometric information to be re-injected at arbitrary depths.

**Integration operators.** We study three principal types of $\mathcal{M}_l$:

1. **Affine Modulation (FiLM) (Perez et al., 2018):** $\mathcal{M}_l(z, m) = \gamma_l(m) \odot z + \beta_l(m)$, where $\beta_l, \gamma_l$ are geometry encoders and $\odot$ is Hadamard product,

2. **Additive injection:** $\mathcal{M}_l(z, m) = z + W_l m$,

3. **Concatenation:** $\mathcal{M}_l(z, m) = \text{Proj}_l([z; m])$.

**Definition 3.5** (Injection policies). Let $S \in \{0, 1\}^L$ be the injection policy vector, where $S_l$ determines whether memory is injected at layer $l$. We consider four structured policies: *Early injection*, defined by $S_l = 1$ for $l \leq \lfloor L/2 \rfloor$; *Late injection*, defined by $S_l = 1$ for $l > \lfloor L/2 \rfloor$; *Single-layer injection*, defined by $S_k = 1$ for a fixed $k \in \{0, \dots, L-1\}$; and *Full injection*, where $S_l \equiv 1$ for all $l$.

We highlight that our memory injection strategy only add constant memory and computation cost for any architecture (in the memory encoder). Thus, any architecture can try to utilize our proposed memory injection to boost its performance.

## 4. Experimental Results

We now provide extensive empirical evidence validating the proposed framework across multiple settings. All our experiments is done on one NVIDIA 5090 GPU. Our code is available at https://github.com/xym331/neural-operators-forget-geometry.

### 4.1. Dataset Description

We evaluate our framework on three diverse benchmarks designed to stress-test geometric generalisation in fluid dynamics. FlowBench (Tali et al., 2024) serves as our primary testbed for zero-shot geometric adaptation, where we used Lid Driven Cavity subset governed either purely by the Navier-Stokes equations (**LDC-NS**) or by heat transfer in addition to fluid dynamics (**LDC-NSHT**). Critically, we enforce a generalization test by training on one geometry category and evaluating on an entirely distinct, unseen geometry category, assessing the model's ability to extrapolate rather than memorize. This is complemented by **AirfRANS** (Bonnet et al., 2022), which models external Reynolds-Averaged Navier-Stokes (RANS) aerodynamics over varied NACA airfoils , and **Darcy** (adpated from Zhao et al., 2025), which evaluates geometrical robustness by training on pentagonal

*Table 2.* Relative $L^2$ error comparing operators without geometry memory (**No Memory**) and their best memory-injected variants (**With Memory**). **Injection** denotes the layer(s) where memory is applied (L0–L3: shallow to deep; Early/Late: near input/output; All: every layer). **Gain** reports the relative improvement. Best values are highlighted in green.

| Model | Dataset | No Memory | With Memory | Injection | Gain (%) |
|---|---|---|---|---|---|
| FNO | LDC-NS | 3.61e−1 | 1.21e−1 | Late | 66.6 |
| | LDC-NSHT | 4.11e−1 | 2.39e−1 | All | 41.8 |
| | AirfRANS | 6.13e−2 | 3.69e−2 | L0 | 39.8 |
| | Darcy | 2.09e−1 | 9.74e−2 | L3 | 53.4 |
| Transolver | LDC-NS | 1.72e−1 | 9.19e−2 | All | 46.5 |
| | LDC-NSHT | 2.73e−1 | 2.30e−1 | All | 15.8 |
| | AirfRANS | 3.28e−2 | 1.40e−2 | Early | 57.4 |
| | Darcy | 1.16e−1 | 9.11e−2 | L1 | 21.2 |
| LNO | LDC-NS | 9.25e−2 | 8.39e−2 | L3 | 9.23 |
| | LDC-NSHT | 2.36e−1 | 2.29e−1 | L3 | 2.97 |
| | AirfRANS | 1.32e−2 | 9.73e−3 | Late | 26.3 |
| | Darcy | 1.52e−1 | 1.12e−1 | L2 | 26.5 |

and testing on hexagonal and octagonal domains. An illustration of our train-validation-test split for Flowbench and Darcy dataset is shown in Table 1.

*Table 1.* The train-validation-test split in our experiments. Figures of Flowbench dataset is adopted from (Tali et al., 2024) and figures of Darcy dataset is adopted from (Zhao et al., 2025)

| Dataset | Train, Validation | Test |
|---|---|---|
| Flowbench (Tali et al., 2024) |  |  |
| Darcy (Zhao et al., 2025) |  |  |

For complete details on datasets, input/output channel specifications, and training splits, please refer to Appendix A.

## 4.2. Main Results

We begin by empirically validating the *Forgetting Hypothesis* on **FNO** and **Transolver** backbones and demonstrating the efficacy of memory injection in Section 4.2.1. This is followed by a control analysis of the **LNO** in Section 4.2.2, which confirms that architectures with *intrinsic memory* benefit minimally from extrinsic injection. Finally, Section 4.3 presents ablation studies on injection location, encoder architectures, and injection strategies. Comprehensive tabulated results and visual galleries are provided in Appendices B.2 and C, respectively.

### 4.2.1. FNO and Transolver Result

**The Forgetting Hypothesis.** To quantify geometric information loss, we analyzed the layer-wise MSE for geometry mask reconstruction, as shown in Figure 3(a). Visual evidence of this degradation is provided in Figure 3(b), which

displays a representative reconstruction from Transolver's second layer. We complemented these spatial metrics with a spectral frequency analysis in Figure 4 to track boundary information across network depths.[1]

This empirical analysis reveals that the *Forgetting Hypothesis* manifests through fundamentally different mechanisms in Fourier versus attention-based architectures. The memory-injected FNO exhibits spectral peaks at wavenumbers $k \in \{16, 32, 64\}$. Given the spatial resolution $S = 128$, these wavenumbers correspond to $S/8, S/4, S/2$, representing spatial wavelengths of 8, 4, 2 pixels that correspond to boundaries of different 'sharpness'. Memory injection helps FNO to preserve these *boundary* information. In contrast, Transolver exhibits *macroscopic* geometric collapse: while its spectral domain signal pattern remain unchanged upon memory injection, it loses the structural context of the domain itself, failing to reconstruct the macroscopic geometry (mask) at deeper layers. [2]

**The Benefit of Memory Injection.** In Table 2, architectures with memory injection consistently outperform models without memory injection. Representative experiments of FNO and Transolver on LDC-NSHT dataset is shown in Figure 2.

### 4.2.2. INTRINSIC VS. EXTRINSIC MEMORY: THE CASE OF THE LAPLACE NEURAL OPERATOR

To validate that the performance gains in FNO and Transolver stem specifically from restoring lost geometric information, we investigate the LNO as a control case. Unlike Fourier-based methods, which implicitly assume periodicity, LNO utilizes the Laplace transform, which can effectively capture transient as well as periodic dynamics. This correspond to boundary effect in spatial domain and prevents the 'washing out' of boundary information observed in FNO as well as the 'forgetting' of macroscopic geometry information observed in Transolver. We termed this property of LNO *implicit memory*.

Our experiments in Figure 5 strongly corroborate this theoretical analysis. Layer-wise geometric reconstruction (Figure 5(a)) confirms that the standard LNO backbone naturally retains macroscopic geometry information across depth, rendering the extrinsic 'reminder' redundant. Similarly, spectral frequency analysis (Figure 5(b)) reveals that standard

---

[1]The final layer is excluded as it immediately precedes the output projection. At this final stage, the feature transitions from latent representations to approximating the target solution field, rendering geometric probing less indicative of the internal memory dynamics observed in first 3 layers.

[2]We posit that for FNO, retaining spectral boundary information in latent space is a necessary condition for outputs' geometry awareness. For Transolver, however, the final layer is capable of recovering sharp boundaries from latent features that appear spectrally smooth.

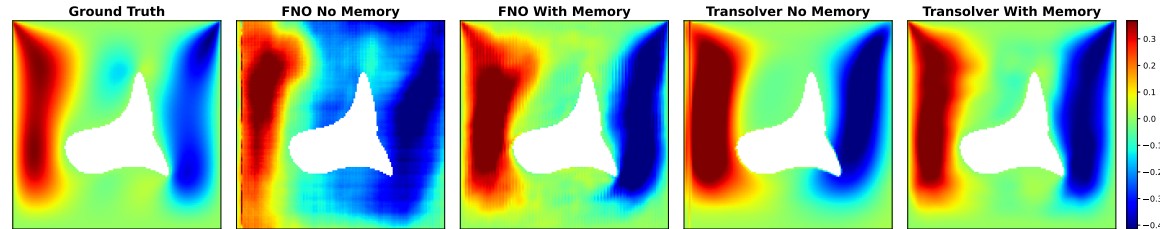

*Figure 2.* Impact of Memory Injection (LDC-NSHT). Without memory, FNO loses flow dynamics and Transolver ignores the obstacle. Injecting memory at all layers correct these failure modes.

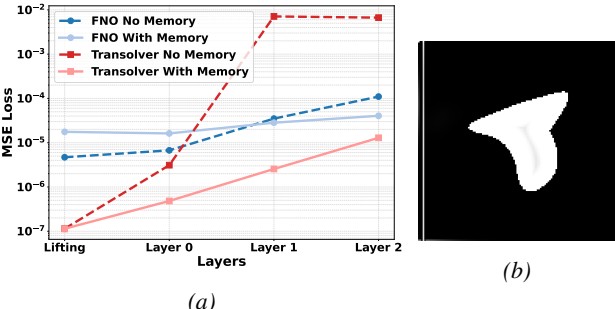

*Figure 3.* Validation of the Forgetting Hypotheses (LDC-NSHT). (a) Reconstruction MSE of the domain mask from hidden states. Numbers in this figure available in Appendix B.1, (b) Representative reconstruction from the Layer 2 hidden state of a standard Transolver, note the blurring in the middle.

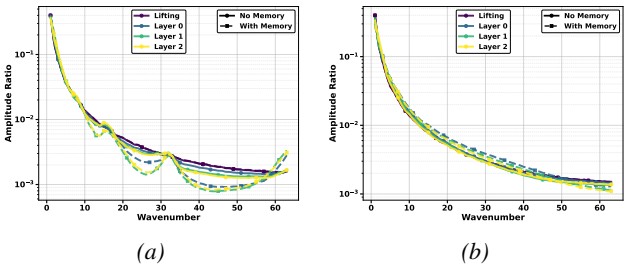

*Figure 4.* Spectral frequency analysis on the LDC-NSHT dataset. Memory is injected at all layers for 'With Memory' model.(a) Memory injection helps FNO retain boundary information, (b) Memory injection does not alter Transolver's spectral domain signal pattern.

LNO (solid line) already has spectral peaks at 16, 32, 64 $(S/2, S/4, S/8$ given resolution $S = 128)$ which is indicative of boundary information. In contrast to FNO, LNO does not need explicit memory injection to preserve these boundary information.

This observation is further supported by our experiments on memory injection in Table 2. The standard LNO baseline achieves high-fidelity results that significantly outperform the standard FNO baseline without requiring any external intervention. Unlike the FNO and Transolver architectures—where memory injection is a decisive factor in reducing error—the LNO model sees negligible perfor-

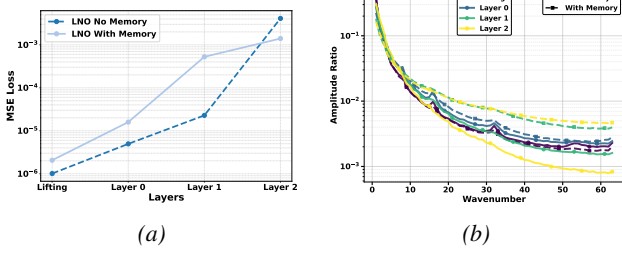

*Figure 5.* LNO Control Analysis (on LDC-NSHT). Unlike FNO or Transolver, memory injection yields minimal gain in (a) reconstruction MSE or (b) boundary information. This negative result confirms LNO possesses intrinsic memory.

mance gains, and in some cases, marginal degradation (see Table 8), when subjected to the same injection protocol. This suggests that the geometric information our method attempts to restore is already inherently present within the LNO architecture.

This 'negative' result serves as a powerful validation of the *Forgetting Hypothesis*. It demonstrates that our method is not merely a generic performance booster; rather, it specifically targets the loss of geometric information, either macroscopic geometry information or boundary information. Since LNO naturally preserves both via its pole-residue formulation, it requires no extrinsic restoration, confirming that the vulnerability observed in FNO and Transolver is indeed a specific deficiency in geometry information retention.

Here we use 'implicit memory' to highlight LNO's capability at dealing with boundary information (and thus irregular geometry) due to its pole-residue formulation. LNO has 'intrinsic memory' does **not** means the tructure of LNO break the Markov chain structure of layered neural operators. As shown in Table 2, even LNO designed to deal with boundary information benefit from our memory injection framework.

### 4.3. Ablation Study and Analysis

We conducted comprehensive ablation studies to evaluate the impact of three key design components: memory injection location, encoder architecture, and injection strategy, whose specific configurations are summarized in Table 3.

*Table 3.* Summary of ablation study. Ablation done is **bold faced**.

| Section | Ablation on | Injection Location | Encoder | Injection Strategy |
|---|---|---|---|---|
| 4.3.1 | Injection Location | **0,1,2,3** | U-Net | FiLM |
| 4.3.2 | Encoder | All | **U-Net, DeepONet** | FiLM |
| 4.3.3 | Injection Strategy | Early,Late, All,0,1,2,3 | U-Net | **FiLM, Concatenation, Additive injection** |

*Table 4.* Ablation of geometry memory injection depth (relative $L^2$ error). **No Memory** denotes the baseline. Highlighted cells indicate catastrophic degradation under final-layer injection, evidencing the *Geometric Shortcut*.

| Model | Injection | LDC-NS | LDC-NSHT | AirfRANS | Darcy |
|---|---|---|---|---|---|
| FNO | No Memory | 3.61e−1 | 4.11e−1 | 6.13e−2 | 2.09e−1 |
| | L0 | 1.33e−1 | 2.52e−1 | 3.69e−2 | 1.14e−1 |
| | L1 | 1.28e−1 | 2.49e−1 | 4.16e−2 | 1.10e−1 |
| | L2 | 1.22e−1 | 2.52e−1 | 4.00e−2 | 9.87e−2 |
| | L3 | 1.22e−1 | 2.46e−1 | 4.47e−2 | 9.74e−2 |
| Transolver | No Memory | 1.72e−1 | 2.73e−1 | 3.28e−2 | 1.16e−1 |
| | L0 | 1.01e−1 | 2.42e−1 | 1.67e−2 | 1.09e−1 |
| | L1 | 1.03e−1 | 2.37e−1 | 1.66e−2 | 9.11e−2 |
| | L2 | 1.11e−1 | 2.30e−1 | 1.67e−2 | 1.11e−1 |
| | L3 | 5.48e−1 | 4.95e−1 | 3.57e−2 | 9.11e−2 |

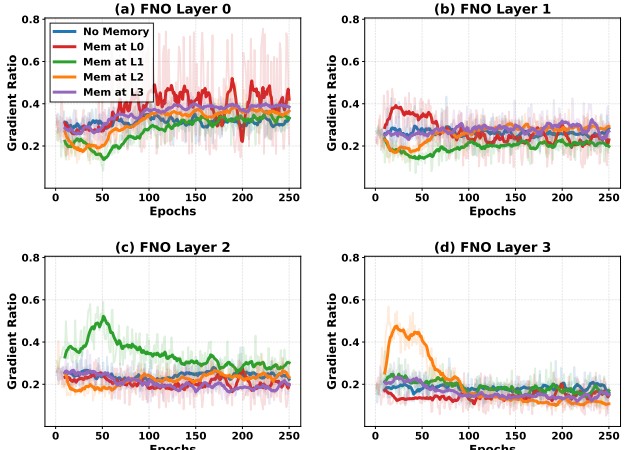

*Figure 6.* FNO Layer-wise Gradient Ratios (LDC-NS). Evolution of relative gradient magnitude for different memory injection locations (Mem at L0–3). The network retains significant gradient signals in early layers even when memory is injected at the final layer, indicating stable learning dynamics.

### 4.3.1. OPTIMIZATION DYNAMICS: ROBUSTNESS VS. SHORTCUT LEARNING

A critical finding of our ablation study is the divergence in stability between Attention-based and Fourier-based architectures under late memory injection. While FNO remains robust regardless of injection depth, Transolver suffers a catastrophic failure when geometry is injected at the final

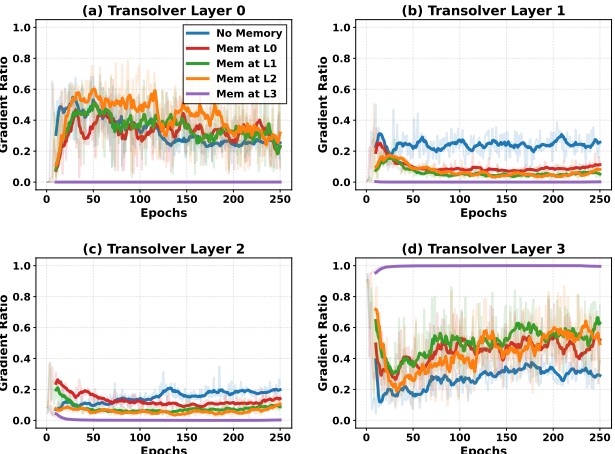

*Figure 7.* Evidence of the *Geometric Shortcut* in Transolver. Under late injection (Mem at L3), the gradient ratio saturates to 1.0 at the final layer (d) while collapsing to 0.0 in all upstream layers (a–c). The optimizer exploits the injected geometry to bypass the physics backbone, leading to feature collapse.

layer.

As shown in Table 4, when memory is injected into Transolver at the final layer (Layer 3), the test error on LDC-NS, LDC-NSHT, AirfRANS explodes. In stark contrast, FNO benefits from memory injection even at the final layer.

To understand the root cause of Transolver's collapse, we analyzed the *Gradient Ratio*—the relative magnitude of gradients at layer $l$ compared to the total network gradient ($R_l = ||\nabla_{\theta_l}\mathcal{L}|| / \sum_k ||\nabla_{\theta_k}\mathcal{L}||$). This metric reveals which layers drive the learning process.

Figure 7 illustrates a phenomenon we term **Geometric Shortcut**: In Layer 3 Injection (Purple line), the gradient ratio for the final layer saturates to 1.0, while the ratios for all preceding backbone layers collapse to 0.0.

Because the memory injection occurs *after* Layer 3 ($v_{\text{out}} = v_3 \odot \gamma + \beta$), the optimizer discovers a greedy solution: it relies primarily on the geometric embeddings ($\gamma, \beta$) from the Memory Encoder. Consequently, Layer 3 is repurposed from a physics processor into a 'carrier layer', adjusting its weights solely to act as a compatible substrate for the geometry injection. This greedy adaptation at Layer 3 absorbs the majority of the gradient signal, effectively shielding the upstream layers (0, 1, 2) from learning. The network 'shortcuts' the global physic modeling (which requires depth) in favor of a superficial fit at the final interface.

We further test Transolver on Darcy dataset with only 1 layer and memory injection, whose error is 1.11e−1. This confirms that Transolver's success on Darcy dataset upon final layer injection originates from the simplicity of Darcy dataset.

*Table 5.* Ablation of the geometry memory encoder (relative $L^2$ error). Both U-Net and DeepONet encoders consistently improve performance over **No Memory**, showing that gains arise from geometric restoration rather than encoder choice. Best values per dataset are highlighted in green.

| Model | Encoder | LDC-NS | LDC-NSHT | AirfRANS | Darcy |
|---|---|---|---|---|---|
| FNO | No Memory | 3.61e−1 | 4.11e−1 | 6.13e−2 | 2.09e−1 |
| | U-Net | 1.16e−1 | 2.39e−1 | 4.27e−2 | 1.25e−1 |
| | DeepONet | 1.69e−1 | 2.94e−1 | 5.00e−2 | 1.68e−1 |
| Transolver | No Memory | 1.72e−1 | 2.73e−1 | 3.28e−2 | 1.16e−1 |
| | U-Net | 9.19e−2 | 2.30e−1 | 1.51e−2 | 9.61e−2 |
| | DeepONet | 1.46e−1 | 2.57e−1 | 1.85e−2 | 9.52e−2 |

*Table 6.* Injection mechanism ablation on LDC-NS (relative $L^2$ error). FiLM remains stable under late injection, whereas additive mechanisms (**Add**itive injection, **Concat**enation) lead to severe degradation in Transolver.

| Model | Injection | FiLM | Add | Concat |
|---|---|---|---|---|
| FNO No Mem: 3.62e−1 | Early | 1.29e−1 | 1.49e−1 | 1.70e−1 |
| | Late | 1.21e−1 | 1.46e−1 | 1.58e−1 |
| | L3 | 1.22e−1 | 1.81e−1 | 1.44e−1 |
| Transolver No Mem: 1.72e−1 | Early | 9.74e−2 | 1.27e−1 | 1.23e−1 |
| | Late | 1.03e−1 | 2.83e−1 | 5.20e−1 |
| | L3 | 5.48e−1 | 5.84e−1 | 6.91e−1 |

FNO (Figure 6) exhibits remarkable stability. Even with final layer injection, the gradient ratio at the final layer remains moderate, allowing significant gradient signal to propagate back to the lifting and initial spectral layers.

This analysis proves that *Early Injection* is not merely an architectural preference for Transolver but a stability requirement. It forces the network to integrate geometric constraints during feature construction, preventing the optimizer from exploiting the geometry shortcut.

#### 4.3.2. TYPES OF MEMORY ENCODER

To ensure that the observed performance gains are not an artifact of the specific U-Net architecture used for the geometry encoder, we conducted a comparative ablation using a DeepONet-based encoder. This setup tests the universality of the *Forgetting Hypothesis* by isolating the injection mechanism from the encoding architecture.

As summarized in Table 5, the benefits of *Geometry Memory Injection* remain consistent regardless of the encoder employed. While the U-Net encoder generally achieves lower absolute errors, the DeepONet-based memory injection still yields substantial improvements over the baseline 'No Memory' models. These results strongly support the conclusion that the performance bottleneck is indeed the loss of geometry information in deep layers. The explicit re-injection of geometric information is the critical remedial factor, irrespective of the specific architecture used to encode those constraints.

#### 4.3.3. TYPES OF INJECTION STRATEGY

We evaluated three distinct injection mechanisms—FiLM, Concatenation, and Additive injection mentioned in Section 3.4. As shown in Table 6, optimal explicit memory injection consistently improves performance over the baseline, regardless of the specific strategy employed. However, the choice of mechanism reveals a fundamental stability gap between spectral and attention-based architectures.

While FNO proves robust to all injection methods, Transolver exhibits a critical vulnerability to Concatenation and Additive injection. Under Late and final layer injection location, these methods fail to prevent the *Geometric Shortcut*, leading to geometric shortcut. In contrast, FiLM partly mitigates this instability.

## 5. Conclusion

In this work, we formalize and validate the **Geometric Forgetting Hypothesis**: the observation that the global mixing mechanisms inherent to deep neural operators—specifically FFT and Self-Attention—progressively attenuate geometry information as network depth increases. In particular, attention based operator experience *macroscopic* geometry forgetting while Fourier based operators struggle with retaining *boundary* information. We utilize **Geometry Memory Injection** as a remedy of this forgetting instead of merely as a performance booster.

Beside these, we offered two insights into deep neural operator architectures:

• **Intrinsic vs. Extrinsic Memory:** Our control analysis using the LNO provides a rigorous confirmation of the forgetting phenomenon. Because LNO mathematically encodes boundary conditions via the pole-residue formulation and hence has *intrinsic memory*, it derives minimal benefit from our injection framework. This negative result confirms that the vulnerability observed in standard FNO and Transolver backbones is specific structural deficiencies of forgetting either macroscopic geometry or boundary information, rather than a capacity limitation that can be resolved by merely scaling parameters or increasing depth.

• **The Geometric Shortcut:** We uncovered a critical divergence in how spectral and attention-based architectures react to geometry information re-injection. While FNO remains robust to late-stage injection, Transolver exhibit a **Geometric Shortcut**, where the optimizer exploits explicit geometry information to bypass major physics backbones, leading to feature collapse.

Ultimately, our findings suggest that in deep operator learning, geometry information function as persistent constraints that must be actively maintained against the 'washing out' effect of global spectral or attention layers. We hope this necessitates a shift in research focus: rather than relying on increasingly complex input encoders and using heuristic injection strategies, more attention should be placed on principled architectural mechanisms that actively preserve geometric memory throughout the network depth.

**Limitations and outlooks.** Our study has limitations. First, reliance on Cartesian grids necessitates interpolation, potentially introducing aliasing artifacts near complex boundaries. Second, our evaluation is restricted to steady-state problems; extending this to time-dependent dynamics is a critical future direction. Finally, while we provide strong empirical validation, establishing a rigorous theoretical quantification of geometric forgetting—perhaps leveraging Information Bottleneck Theory (Tishby & Zaslavsky, 2015)—remains an open challenge. This would require rigorous description and parametrization of input output as random variables. Moreover, as a simple remedy like the memory injection we proposed works well, we expect more sophisticated memory mixing mechanism, *e.g.*, LSTM, would boost neural operators performance on irregular geometry in a direction orthogonal to designing complicate geometry encoders.

## Impact Statement

This paper presents work whose goal is to advance the field of Machine Learning. There are many potential societal consequences of our work, none which we feel must be specifically highlighted here.

## Acknowledgement

We thank Chun-Wun Cheng for helpful discussions. AIAR gratefully acknowledges the support of the YMSC, Tsinghua University. This work is also supported by Tsinghua University Initiative Scientific Research Program, and Tsinghua University Dushi Program.

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

# A. Detailed Dataset Description

## A.1. Flowbench-LDC

We utilize the 2D Lid-Driven Cavity subsets from the FlowBench dataset. These provide high-fidelity steady-state solutions for incompressible flows within a cavity containing complex internal obstacles. We consider two distinct regimes: (1) LDC-NS, governed purely by the Navier-Stokes equations, and (2) LDC-NSHT, a multiphysics problem coupling fluid dynamics with heat transfer. Critically, we assess geometric generalization by training on one class of shapes ('G2') and testing on a distinct class ('G1'). See Figure 8.

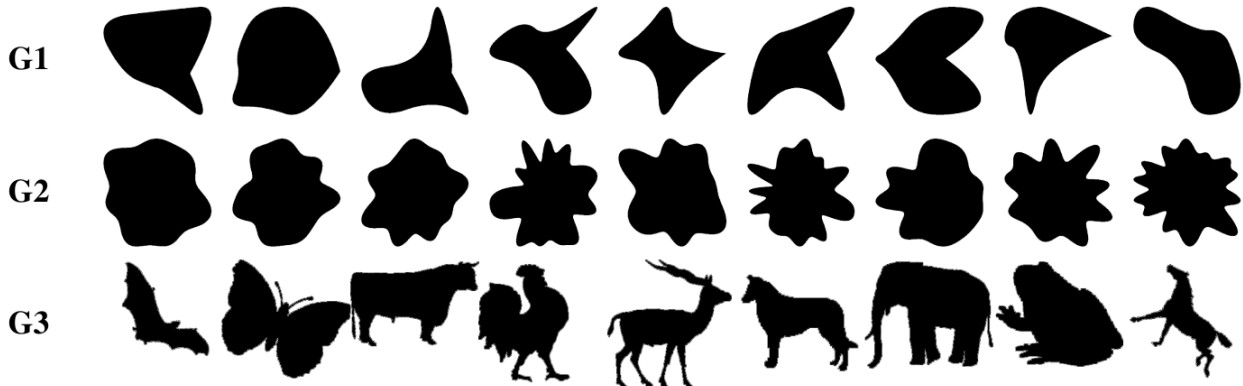

*Figure 8.* Flowbench dataset shapes. This Figure is adopted from (Tali et al., 2024)

**Inputs and Outputs**

- **LDC-NS**: The input tuple consists of $(Re, \mathrm{SDF}, \mathrm{Mask}, \mathbf{x})$, where $Re$ is the Reynolds number, and $\mathbf{x}$ are coordinate channels. The output is the velocity field $(u_x, u_y)$ and pressure $p$.

- **LDC-NSHT**: The input tuple is expanded to include the Richardson number ($Ri$), representing the ratio of buoyancy to inertial forces: $(Re, Ri, \mathrm{SDF}, \mathrm{Mask}, \mathbf{x})$. The output includes temperature $T$ in addition to velocity: $(u_x, u_y, T)$.

**Splits and Generalization**    To rigorously test geometric memory, we enforce a strict separation of shape categories.

- **Training**: 800 samples from the 'G2' category.

- **Validation**: 200 samples from the 'G2' category.

- **Testing**: 800 samples from the 'G1' category, testing zero-shot generalization to unseen shapes.

## A.2. AirfRANs

This dataset captures steady-state aerodynamics over NACA airfoils in a subsonic regime, governed by the Reynolds-Averaged Navier-Stokes (RANS) equations. It represents an external flow problem where the geometry (the airfoil) is defined by a mask and Signed Distance Function (SDF). We interpolate the original unstructured mesh data onto a regular grid to facilitate standard neural operator processing.

**Preprocessing**    The original dataset is defined on unstructured meshes. We interpolated these fields onto a regular Cartesian grid using nearest-neighbor interpolation.

**Inputs and Outputs**

- **Inputs**: Reynolds number ($Re$), Angle of Attack ($\alpha$), SDF, Mask, and coordinates $\mathbf{x}$.

- **Outputs**: Steady-state velocity components $(u_x, u_y)$ and pressure $p$.

**Splits**    The dataset of 1,000 simulations is split into 600 training, 200 validation, and 200 test samples.

### A.3. Darcy

We adopt the Darcy flow benchmark from (Zhao et al., 2025), which models diffusion in porous media. This dataset challenges the model to generalize across domain geometries: training is performed on irregular pentagonal domains, while testing is conducted on hexagonal and octagonal domains. The original dataset is defined on unstructured meshes of query points. We interpolated these fields onto a regular Cartesian grid using nearest-neighbor interpolation.

**Training Domain.**   Irregular pentagons.

**Testing Domain.**   Hexagons and Octagons. This evaluates the model's ability to handle changes in domain geometry.

**Tasks.** Map the diffusion coefficient $a(x, y)$ (permeability) to the pressure solution $u(x, y)$.

## B. Raw Data

### B.1. The Forgetting Test

Our lightweight decoder consist of Conv2d $\rightarrow$ GELU $\rightarrow$ Conv2d $\rightarrow$ Sigmoid. The numerical result for the forgetting test is in Table 7.

*Table 7.* Forgetting Test on LDC-NSHT dataset. Here memory is injected at all layers for 'With Memory' models. Transolver benefit most from memory injection, while FNO and LNO benefit marginally, sometimes even suffer degradation, upon memory injection

|  |  | Lifting | Layer 0 | Layer 1 | Layer 2 |
|---|---|---|---|---|---|
| FNO | No Memory | 4.69e−6 | 6.72e−6 | 3.50e−5 | 1.10e−4 |
|  | With Memory | 1.76e−5 | 1.62e−5 | 2.83e−5 | 4.05e−5 |
| Transolver | No Memory | 1.14e−7 | 3.11e−6 | 7.11e−3 | 6.67e−3 |
|  | With Memory | 1.14e−7 | 4.83e−7 | 2.55e−6 | 1.29e−5 |
| LNO | No Memory | 1.01e−6 | 4.95e−6 | 2.28e−5 | 4.12e−3 |
|  | With Memory | 2.06e−6 | 1.59e−5 | 5.26e−4 | 1.40e−3 |

## B.2. Geometry Memory Injection

*Table 8.* Summary of all result using FiLM injection. Best result for each model for each dataset are highlighted in green. Degradation from No Memory model are highlighted in red.

| Model | Location | LDC-NS | LDC-NSHT | AirfRANS | Darcy |
|---|---|---|---|---|---|
| FNO | Non | 0.36148 | 0.41060 | 0.06126 | 0.20908 |
| | Early | 0.12915 | 0.25293 | 0.04397 | 0.10824 |
| | Late | 0.12080 | 0.24875 | 0.03824 | 0.11165 |
| | All | 0.11621 | 0.23882 | 0.04269 | 0.12542 |
| | L0 | 0.13338 | 0.25202 | 0.03687 | 0.11400 |
| | L1 | 0.12822 | 0.24935 | 0.04156 | 0.11036 |
| | L2 | 0.12204 | 0.25163 | 0.03996 | 0.09867 |
| | L3 | 0.12167 | 0.24645 | 0.04469 | 0.09736 |
| Transolver | Non | 0.17185 | 0.27296 | 0.03282 | 0.11555 |
| | Early | 0.09740 | 0.23175 | 0.01398 | 0.09356 |
| | Late | 0.10314 | 0.39573 | 0.01814 | 0.11111 |
| | All | 0.09188 | 0.22983 | 0.01507 | 0.09607 |
| | L0 | 0.10122 | 0.24216 | 0.01671 | 0.10899 |
| | L1 | 0.10251 | 0.23714 | 0.01661 | 0.09106 |
| | L2 | 0.11076 | 0.23044 | 0.01671 | 0.11123 |
| | L3 | 0.54794 | 0.49497 | 0.03573 | 0.09113 |
| LNO | Non | 0.09248 | 0.23582 | 0.01320 | 0.15179 |
| | Early | 0.08906 | 0.25346 | 0.01029 | 0.14652 |
| | Late | 0.08443 | 0.23401 | 0.00973 | 0.11368 |
| | All | 0.08595 | 0.23111 | 0.01000 | 0.14523 |
| | L0 | 0.08821 | 0.23775 | 0.01215 | 0.12848 |
| | L1 | 0.08974 | 0.22919 | 0.01048 | 0.16756 |
| | L2 | 0.10171 | 0.23000 | 0.01032 | 0.11162 |
| | L3 | 0.08394 | 0.22882 | 0.00981 | 0.13338 |

# C. Gallery

Here we exemplify the benefit of **geometry memory injection** through graphics across different datasets.

## C.1. LDC-NS

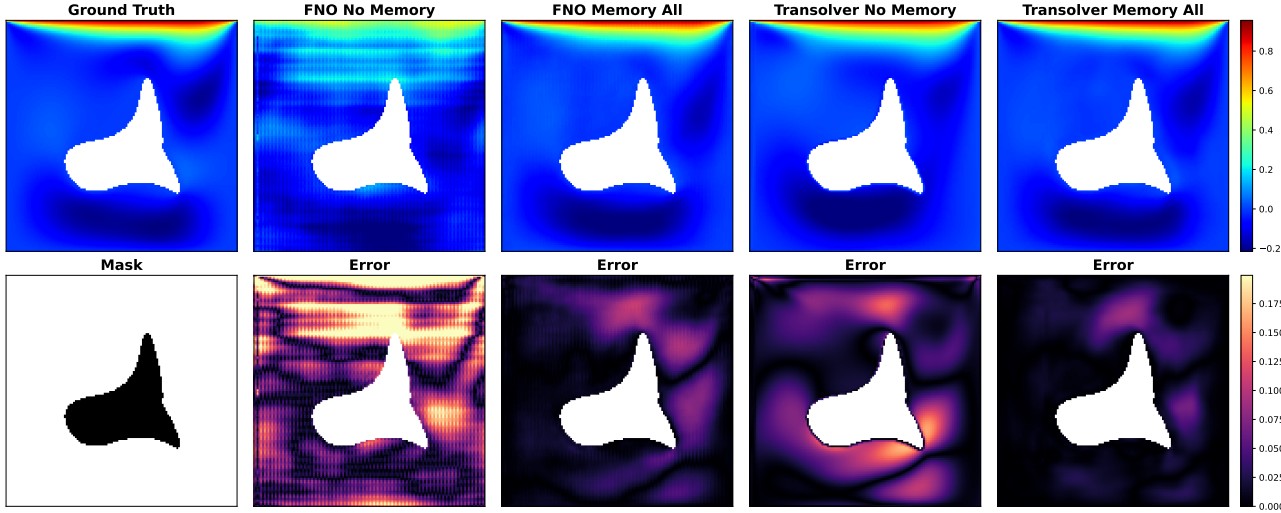

*Figure 9.* Visualization of LDC-NS velocity in horizontal direction

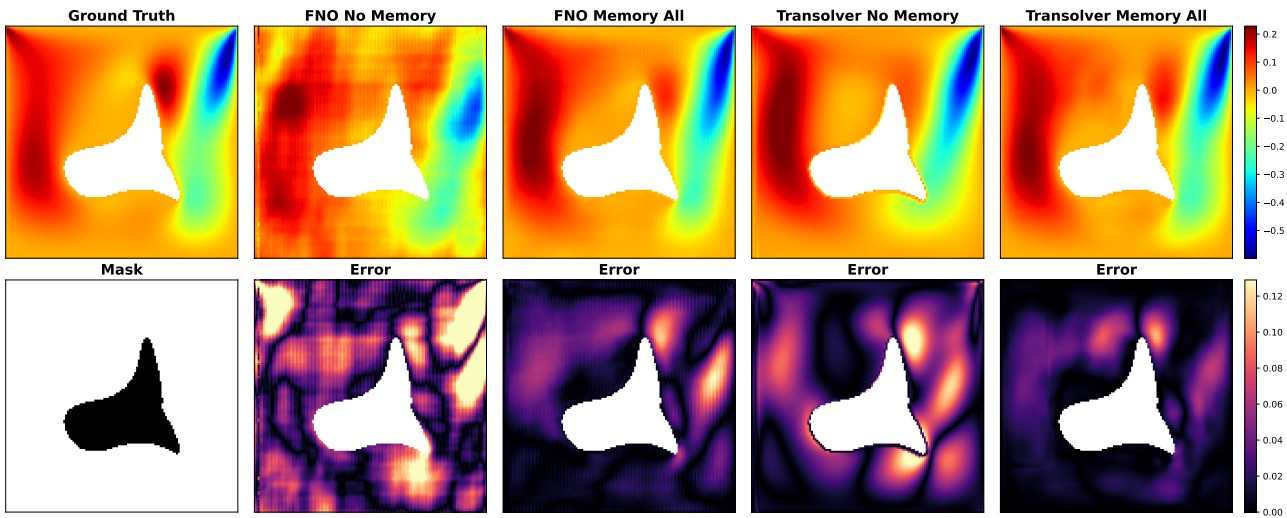

*Figure 10.* Visualization of LDC-NS velocity in vertical direction

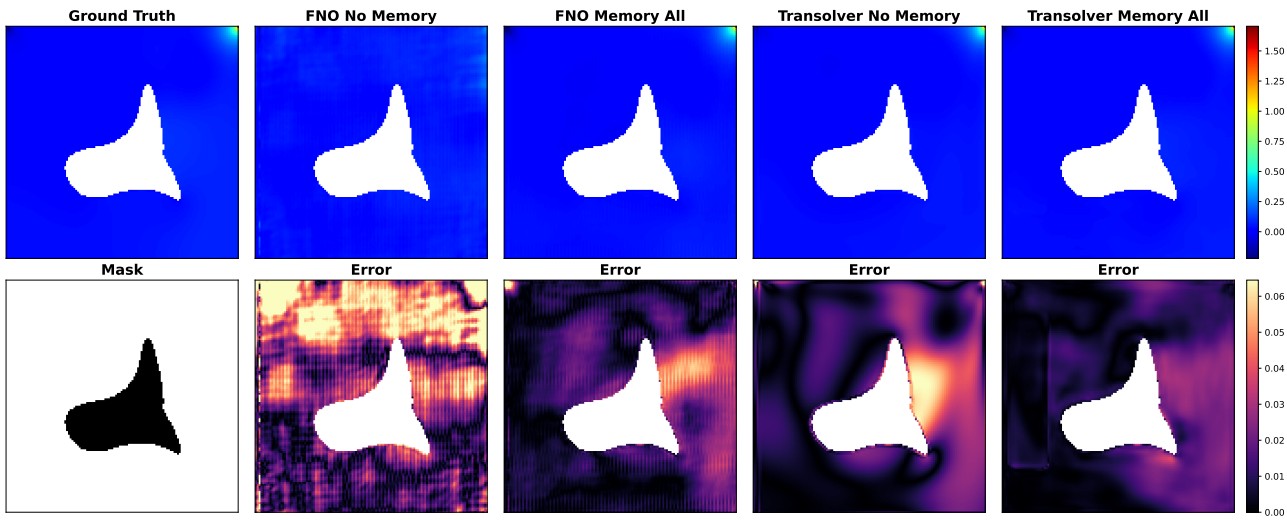

*Figure 11.* Visualization of LDC-NS pressure

## C.2. LDC-NSHT

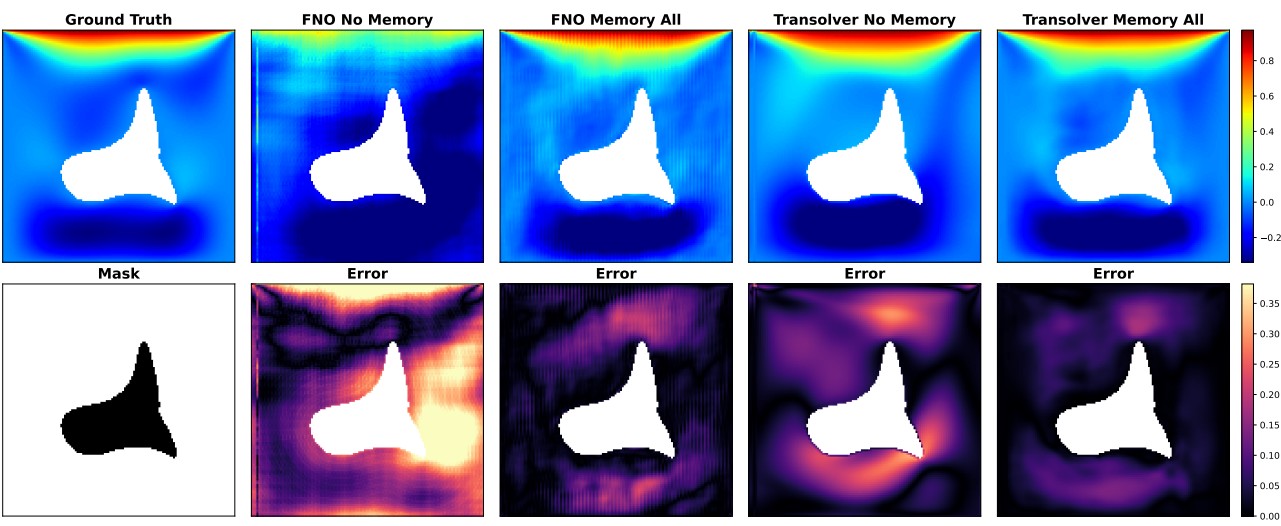

*Figure 12.* Visualization of LDC-NSHT velocity in horizontal direction.

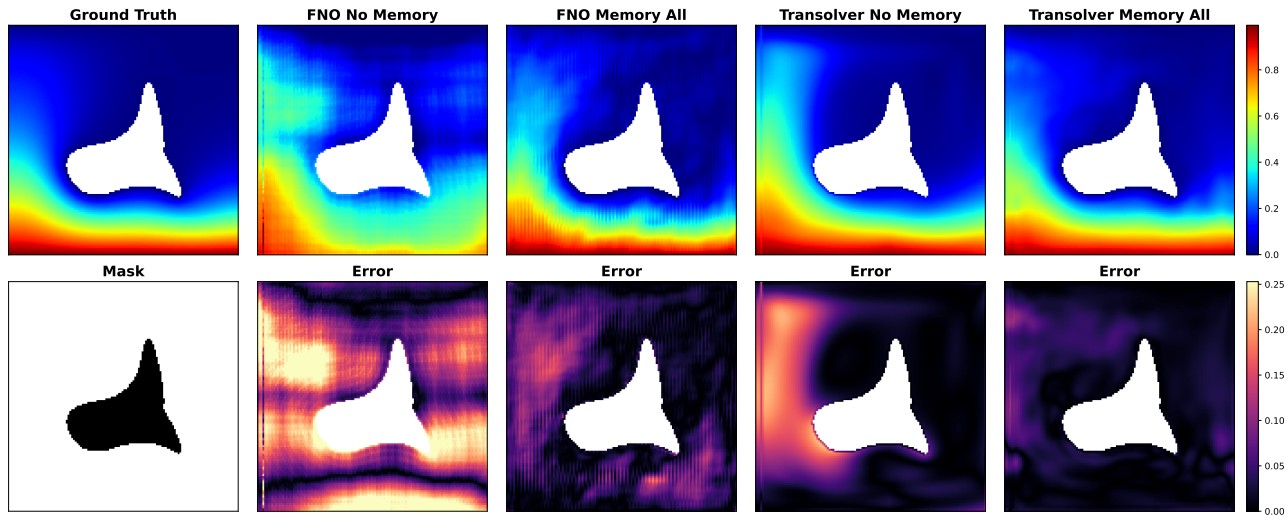

*Figure 13.* Visualization of LDC-NSHT pressure.

## C.3. Darcy

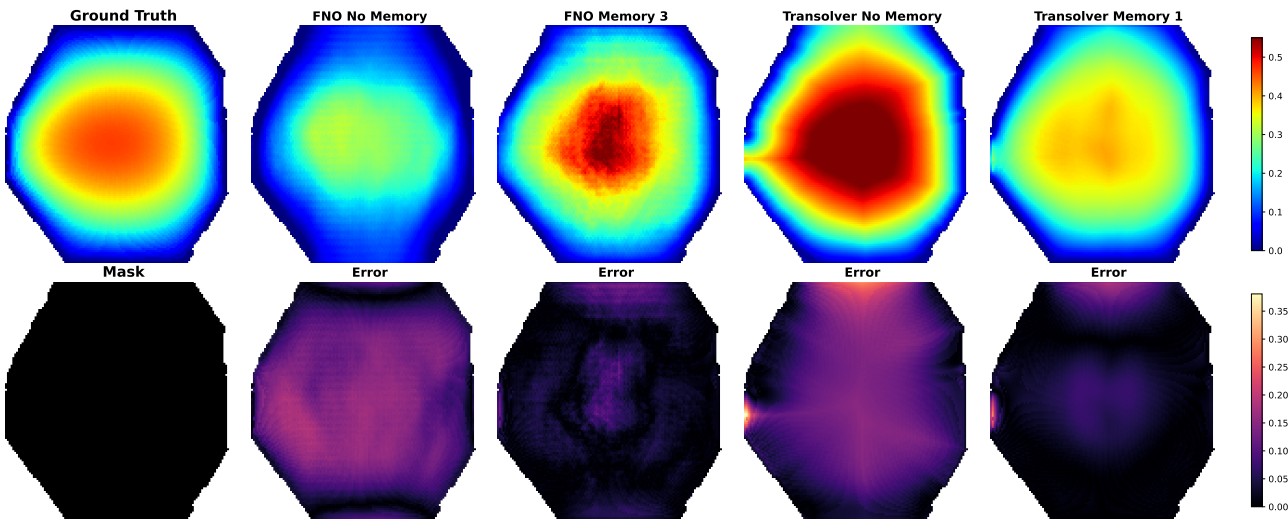

*Figure 14.* Visualization of Darcy result

## C.4. AirfRANS

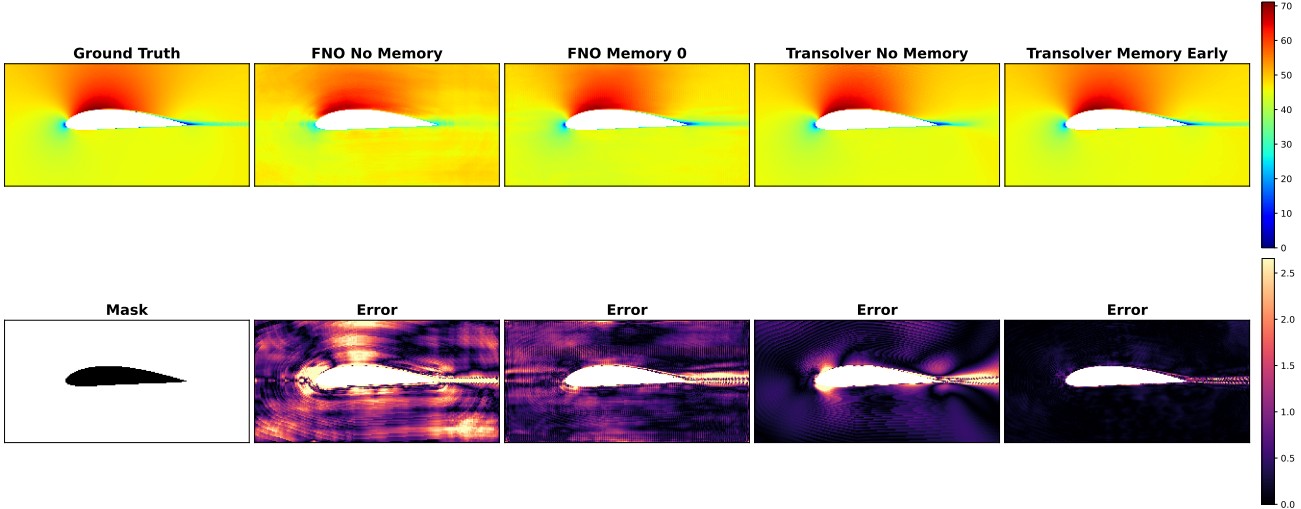

*Figure 15.* Visualization of AirfRANS velocity in horizontal direction

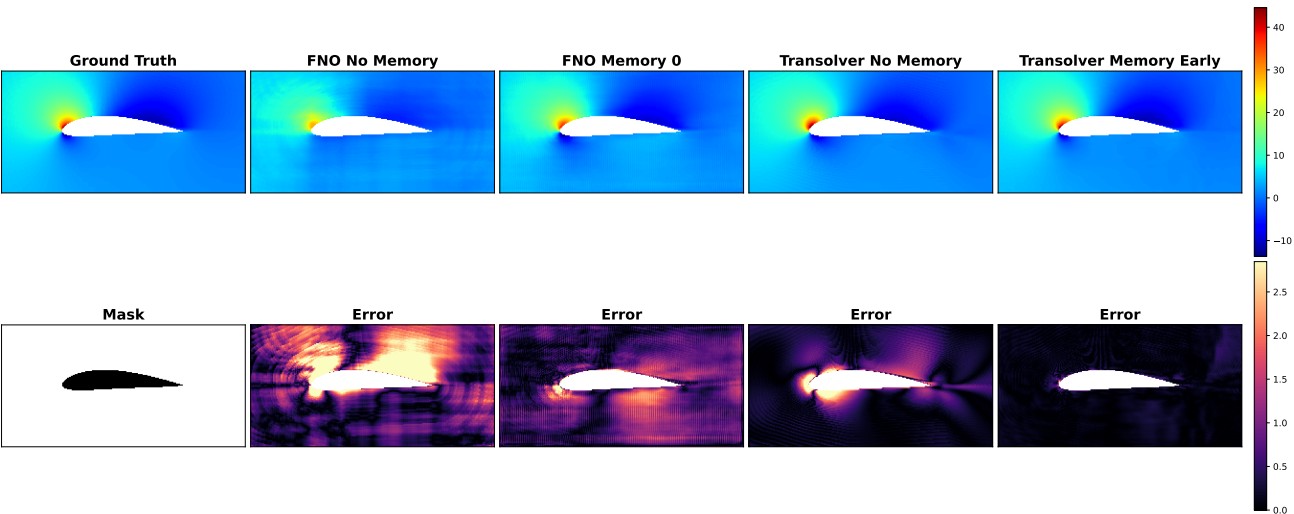

*Figure 16.* Visualization of AirfRANS velocity in vertical direction

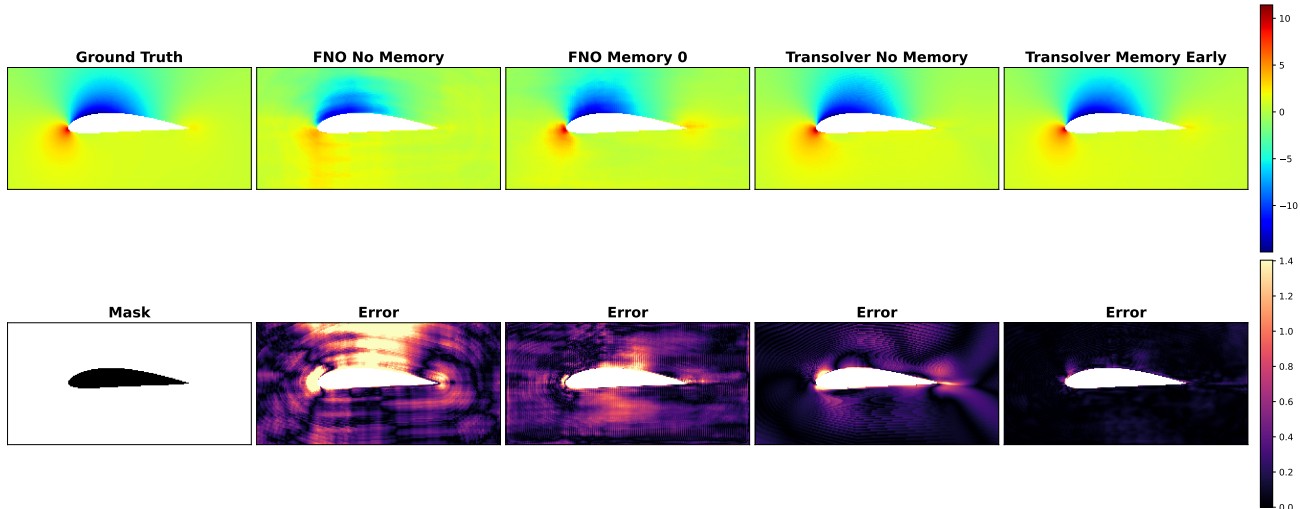

*Figure 17.* Visualization of AirfRANS pressure

