# OpenReview forum: "Do Neural Operators Forget Geometry? The Forgetting Hypothesis in Deep Operator Learning"
_ICML.cc/2026/Conference — ICML 2026 regular_

### Official Review · Reviewer_QeWr · 2026-03-12

**Soundness:** 3
**Presentation:** 3
**Significance:** 4
**Originality:** 4
**Overall Recommendation:** 6
**Confidence:** 3

**Summary:**

This paper identifies and formalises a structural failure mode in deep neural operators: the progressive loss of domain geometry information as representations propagate through depth. The core observation is that standard neural operator architectures (e.g., FNO, Transolver) admit a Markovian layer structure, where geometry G is injected only at the input. Applying the Data Processing Inequality to this chain yields Proposition 3.1, formalising the intuition that geometric information cannot increase with depth and is progressively attenuated by global mixing operations (FFT, self-attention). The authors term this the Geometric Forgetting Hypothesis.

They operationalise the hypothesis via two diagnostic proxies: a layer-wise mask reconstruction error ($epsilon_l$, measuring macroscopic geometry retention) and a spectral power analysis ($rho_{l(kappa)}$, tracking boundary information). They empirically validate both diagnostics on FNO and Transolver across four benchmarks (FlowBench LDC-NS, LDC-NSHT, AirfRANS, Darcy), demonstrating that forgetting manifests differently in the two architectures: FNO loses spectral boundary information, while Transolver loses macroscopic geometry.

To counteract forgetting, the authors introduce Geometry Memory Injection: a mechanism that explicitly reintroduces the geometry encoding G at intermediate layers via modulation operators $M_l$ (FiLM, additive, or concatenation). They compare four injection policies (Early, Late, Single-layer, Full). Memory injection consistently reduces relative L2 error, with gains of 15-67\% on FNO and 15-57\% on Transolver across benchmarks.

A control experiment using the Laplace Neural Operator (LNO) provides a principled negative result: LNO's pole-residue formulation naturally encodes boundary conditions, giving it intrinsic geometric memory. Accordingly, LNO benefits minimally from extrinsic injection, while
its baseline performance already surpasses FNO without memory. This negative result is one of the paper's strongest methodological moves.

A secondary finding is the Geometric Shortcut in Transolver: under late-layer geometry injection, the optimizer bypasses the physics backbone entirely, saturating gradients at the injection layer while collapsing gradients in all upstream layers. This is diagnosed via a layer-wise Gradient Ratio metric and shown to be a Transolver-specific instability not observed in FNO.

**Compliance With Llm Reviewing Policy:**

Affirmed.

**Key Questions For Authors:**

Comparison with coordinate-based conditioning. Many existing architectures (Geo-FNO, Transolver++) already inject coordinate or SDF information at multiple layers as a design choice. The paper's contribution is to provide a principled explanation for why this is necessary. But does the proposed Memory Injection mechanism outperform or merely match the performance of these existing architectures on the tested benchmarks? A comparison with Geo-FNO or Transolver++ on at least one benchmark would clarify whether the proposed framework adds value beyond existing multi-layer conditioning designs.

**Limitations:**

YES

**Strengths And Weaknesses:**

SOUNDNESS

Strengths. Proposition 3.1 is correct. Given the Markov structure $G -> V_l -> V_{l+1}$, the Data Processing Inequality gives $I(G; V_{l+1}) <= I(G; V_l)$ immediately, and Corollary 3.2 follows by induction. This is a clean, valid theoretical foundation. The distinction between macroscopic geometry loss (mask reconstruction) and boundary information loss (spectral analysis) is careful and important; these are genuinely distinct phenomena with different diagnostic signatures, and the paper correctly shows that they manifest differently in FNO versus Transolver.

The LNO control study is the methodological highlight of the paper. By selecting an architecture with known intrinsic memory properties and demonstrating that extrinsic injection yields no significant gain (Table 2, LNO: gains of 2-27% compared to 16-67% for FNO and Transolver), the authors provide a controlled test of their causal claim. This is the kind of evidence that distinguishes a diagnostic framework from a mere performance trick.

The Geometric Shortcut analysis (Section 4.3.1) is the paper's most surprising and practically useful finding. The gradient ratio diagnostic is well-motivated, and Figure 7 shows a crisp result: under L3 injection, Transolver's gradient ratio at L3 saturates to ~1.0 while upstream layers collapse to ~0.0. The 1-layer Transolver on Darcy (error 1.11e-1, matching L3 injection with full depth) is a nice sanity check confirming the shortcut interpretation.


SIGNIFICANCE

This paper makes a genuine conceptual contribution to the neural operator literature. The question "does geometry survive depth?" has not been asked systematically before, and the answer ( that it does
not, for different reasons in spectral vs. attention-based architectures) is both surprising and actionable. The Geometric Shortcut finding is practically important: it explains why existing architectures inject geometry continuously rather than at the input, and it provides a gradient-based diagnostic for identifying the
failure mode.

The injection mechanism itself is simple by design. This is not a new architectural component -- FiLM conditioning is well-established, but the paper's contribution is not the mechanism but the diagnosis that motivates it.
The practical gains (15-67\% reduction in relative L2 error) are substantial and consistent across architectures and datasets.

One caveat regarding significance: the evaluation is restricted to steady-state problems on 2D Cartesian grids (with interpolation to the grid for AirfRANS and Darcy). Neural operators on unstructured meshes and time-dependent PDEs, arguably the harder and more practically relevant settings, are acknowledged as future work but not
evaluated. The generalizability of the Forgetting Hypothesis to graph-based operators (GNO, GINO) and time-dependent settings is unclear, as these architectures typically incorporate geometric information in different ways.


PRESENTATION

The paper is well-written and the narrative is clear. The Markov chain framing in Figure 1 is an effective visual summary of the core argument. The distinction between intrinsic and extrinsic memory (Section 4.2.2) is elegantly motivated and executed. The gradient ratio analysis in Section 4.3.1 is clearly presented with strong supporting figures.

ORIGINALITY

The Geometric Forgetting Hypothesis is genuinely new. While the connection between global mixing and information loss has been noted qualitatively in the neural operator literature (as motivation for geometry conditioning), the present paper is the first to formalise it via the DPI, operationalise it with layer-wise probes, and
distinguish between the macroscopic and spectral failure modes.

---

> ### Author Rebuttal · Authors · 2026-03-31
>
> We thank the reviewer for the positive evaluation and for recognising the novelty and practicality of our approach, consistent with the overall positive assessment across reviewers.
>
> **➡️ GeoFNO experiment and comparison with coordinate-based conditioning:**
>
> We thank the reviwer for the insightful suggestion. Vanilla GeoFNO has coordinate conditioning: $v_{t+1}=\sigma(Wv_t+Kv_t+b)$ where $W$ is channel-wise convolution, $K$ is Fourier convolution and $b$ is a bias term **generated by grid coordinates**. In experiment below, we compare vanilla GeoFNO with memory injection, and with GeoFNO minus coordinate conditioning.
> AirfRANS (steady state)
> | Model | Relative L2 |
> |---|---:|
> | GeoFNO (baseline, 4 layers) | 0.0935 |
> | GeoFNO - coordinate conditioning | 0.1094 |
> | GeoFNO + memory (0,1) | 0.0835 |
> | GeoFNO + memory (2,3) | 0.0894 |
>
> It can be seen that coordinate conditioning indeed improve performance, but memory injection still benefit even with coordinate conditioning.
>
> More broadly, we observe the same behaviour across architectures (FNO, Transolver, LNO, GeoFNO), indicating that the effect is **structural rather than tied to a specific implementation of conditioning**.
>
> **➡️ Graph based models and time dependent problems:**
>
> We thank the reviewer for the kind suggestion to broaden our scope. Sp2GNO is a graph based method that take irregular mesh as input and do graph convolution. CFDBench tube flow is a time dependent problem concern flows in tube of different size. Here are result for these:
>
> AirfRANS (irregular mesh, steady state)
> | Model | Relative L2 |
> |---|---:|
> | GeoFNO (baseline, 4 layers) | 0.0935 |
> | GeoFNO + memory (0,1) | 0.0835 |
> | GeoFNO + memory (2,3) | 0.0894 |
> | Sp2GNO (baseline, 6 layers) | 0.2246 |
> | Sp2GNO + memory (0,1,2) | 0.1912 |
> | Sp2GNO + memory (3,4,5) | 0.1948 |
>
>
> CFDBench tube flow (time-dependent)
> | Model | Relative L2 |
> |---|---:|
> | GeoFNO (baseline, 4 layers) | 0.0882 |
> | GeoFNO + memory (0,1) | 0.0597 |
> | GeoFNO + memory (2,3) | 0.0672 |
>
> Across graph-based operators and time-dependent settings, memory injection consistently improves performance. This indicates that the effect is **not restricted to grid-based or steady-state problems**, but reflects a more general issue of **geometry propagation through depth**.

---

### Official Review · Reviewer_nJuC · 2026-03-21

**Soundness:** 3
**Presentation:** 3
**Significance:** 2
**Originality:** 2
**Overall Recommendation:** 2
**Confidence:** 3

**Summary:**

The paper studies neural operator models for PDEs and argues that deeper layers progressively start “forgetting” the domain geometry. It tests this by attaching small decoders and spectral analyses to hidden layers of FNO, Transolver, and LNO, and shows that geometry becomes hard to recover in FNO and Translover. However, it is preserved in LNO. The remedy is to inject geometry into the intermediate layers to improve accuracy on irregular domains.

The contributions are: an information‑theoretic framing of this “geometric forgetting”. A measure of the “fading” geometry information with the depth. A method for re-injection of the geometry, and the evaluation of their method.

**Compliance With Llm Reviewing Policy:**

Affirmed.

**Final Justification:**

Thank you for the reply, but it reiterates the same arguments. I do not share them. After reconsideration and rethinking, I am lowering my score from 3 to 2. The empirical results: 15-67% error reduction, are indeed solid and potentially useful. However, I do not agree with the claim its about the "geometry". In my option the framing claims more than the evidence supports. The SDF indeed curries geometry, but the DPI is not about geometry. Its general. Measuring mutual information is not a geometric trait. The claim "NO forget geometry" is in my option wrongly stated. It forgets "information" but as this paper tries to convey, is a misinterpretation, at least this is my conclusion given the material. I truly believe the paper has discovered something, but it has not provided the right interpretation. Addressing this is not a minor revision---it requires rethinking what "geometry" means in this context, and whether the phenomenon is geometry-specific at all.

**Key Questions For Authors:**

(1) What I was thinking while reading the paper is the analogy to skip connections. That could be a baseline to measure against. Have you tried it?

(2) I am not an expert on DPI but Proposition 3.1 only shows non-increase. Why did the paper conclude and  claim progressive decay from that?

(3) Finally, there is the LNO exception wrt the FNO. The explanation is the “conjecture” in section 3.3. Could it not be underpinned experimentally? As is it is kind of vague.

**Limitations:**

Some limitations are discussed, and the paper itself acknowledges that the theory remains incomplete. In my view, stronger baseline comparisons, especially to skip-connection-based designs, would help clarify whether the proposed interpretation is the right one.

**Strengths And Weaknesses:**

Soundness:
The technical soundness is overall okay on the empirical paper from what I can judge. I am not researching actively on NO and I am not from the PDE/PINN field but I looked into it for a while as I see there is a lot of connection to the INR methods in CG/CV.

I am familiar with Information Theory in general but I do not study it actively. The math of the MC and DPI look correct from what I can judge, I didn’t check it in full depth. In Section 3.2 the paper formalizes the MC and DP and this part seems technically fine but conceptually “bolted on” relative to the experiments. The empirical parts look solid.

However, I am not fully convinced that the DPI theory used to justify the empirical observations is the right way. The Geometric Forgetting Hypothesis is in my opinion an overselling and overreach argument. It's a weak connection at best, and it is kind of obvious. This is not novel in the CV community and this is why other architectures use skip connections. The original ResNet observed that already.

Presentation: The presentation is good, the paper is clearly written and well-structured.

Significance:
The significance of the results is medium: as mentioned the decay over the depth is not something which is novel or unknown. The theory provided here is not very convincing overall. So while it is interesting to analyze it I am not sure that paper does it at the right level

Originality: The idea of going that way is likely a good one, the technical execution is medium, it works kind of, but the theoretical treatment is in my opinion not established.

Summarized: strength: empirical part is okay, weakness: theory is not well connected and does not support strongly, and the idea/observation of geometry decay is not novel.

---

> ### Author Rebuttal · Authors · 2026-03-31
>
> We thank the reviewer for the careful and insightful review. We also thank the reviewer for agreeing that our 'empirical parts look solid'.
>
> **➡️ Relation with skip connections:**
>
> Thanks for the comment. Standard NO architectures already incorporate ResNet-style sequential skip connections (which is $v_{t+1}=\sigma(v_t+\mathcal{F}(v_t))$). For instance, the standard FNO update is $v_{t+1}=\sigma(Wv_t+Kv_t)$, where $K$ is Fourier convolution and $W$ is channel-wise linear transform that acts as a skip connection. Transolver blocks employ similar mechanisms. Because these sequential skip connections are already present in our baselines, they are evidently insufficient to prevent geometric forgetting over many layers.
>
> Our proposed mechanism fundamentally differs from standard skip connections by providing direct memory injection from the **initial** input geometry straight into deep latent representations, bypassing the progressive degradation of layer-by-layer propagation.
>
> **➡️ On DPI and Information Decay**
>
> We appreciate the reviewer’s precision regarding Proposition 3.1. While the DPI  implies a "non-increase," we clarify that in the context of deep neural operators, this upper bound results in as **progressive decay** due to the specific inductive biases of global mixing layers (FFT/Attention). These layers act as **stochastic information sinks**; unless a layer is a perfect identity mapping—which is not the case in optimised networks—the mutual information $I(X; Y_n)$  dissipates as depth increases.
>
> We use mask reconstruction as an empirical proxy for conditional entropy $H(X|Y)$ where $X$ denotes input and $Y$ denote latent representations. Since $I(X;Y) = H(X) - H(X|Y)$, and $H(X)$ is a fixed property of the input distribution, a rising reconstruction error (increased $H(X|Y)$) directly signifies a loss of mutual information. Our experiments confirm this monotonic dissipation across all tested backbones.
>
> We would like to give a further comment on information theoretical analysis. Consider Markov chain $X-Y_1-Y_2-Y_3-Y_4$, where $X$ denotes input variable and $Y_i, i=1,2,3,4$ denote latent representations. Assume each variable is Gaussian centered at previous one and with fixed noise: $Y_1\sim \mathcal{N}(X;\epsilon)$, $Y_i\sim \mathcal{N}(Y_{i-1};\epsilon),i=2,3,4$. Denote mutual information of this model between $X$ and $Y_4$ as $I_1(X;Y_4)$. Now consider another model, where latent representations are denoted by $Z_i,i=2,3,4$ and $Z_1\sim \mathcal{N}(X;\epsilon)$, $Z_i\sim \mathcal{N}(Z_{i-1};\epsilon),i=2,3$, $Z_4\sim t\mathcal{N}(Z_{3};\epsilon)+(1-t)\mathcal{N}(X;\epsilon)$. That is, there is a direct re-injectoin from input to latent representation $Z_4$. Denote mutual informatoin of this model between $X$ and $Z_4$ as $I_2(X;Z_4)$. Then it can be shown (just plug in definition) that there exist a range of $t$ such that $I_2(X;Z_4)>I_1(X;Y_4)$, which means that adding memory injection increase mutual information.
>
> Experimentally, in our experiment in Section 4.3.1 (Table 4), when memory is injected solely at final layer (layer3) in Transolver, Transolver takes a shortcut path that bypass layer0-2, indicating that the short path of re-injection (the $Z_i$ model above) is ideed taken. This confirms that the model actively utilises the re-injected geometry information which was lost during the preceding transformations. Our contribution is therefore not just a theoretical observation of DPI, but the  **high-fidelity information highway** that bypasses the structural decay inherent in deep operator architectures.
>
>
>
> **➡️ Analysis of LNO experiment:**
>
>  This point is supported empirically.
>
> LNO exhibits clear boundary-sensitive structure (e.g., spectral peaks, Fig. 5b), indicating stronger retention of geometric information, whereas FNO lacks such features (Fig. 4a). This aligns with LNO’s pole–residue formulation, which implicitly preserves boundary responses.
>
> However, even in LNO, memory injection yields improvements (Table 8). If geometry were fully preserved, re-injection would be redundant or harmful. Instead, the observed gains indicate that **implicit mechanisms are not sufficient to preserve geometry through depth**.
>
> Thus, the explanation is not only conceptual but **experimentally supported**: LNO retains more geometry than FNO, yet still benefits from explicit conditioning.

---

> > ### Author Rebuttal · Reviewer_nJuC · 2026-04-05
> >
> > Thank you for the rebuttal and added information. However, my concerns were not really resolved or addressed. My major concern is that the claim is too strong: "NO forget geometry". This is too general, and the proposed fix is not really a strong support for either: that the claim is true, nor that it is fixed. The paper shows the decoder cannot recover the exact geometry alone from the deeper layers. Okay, but maybe the geometry is just differently encoded and not "forgotten"? That is not shown. The solution is to inject the input at other deeper layers: this is reasonable and has effects, but moderate at best in my opinion. The concept of doing it that or similar way is essentially almost as old as neural nets are. FiLM is doing it, skip-connections are doing it. Its not really a novelty.
> >
> > The argumentation also does not really reason with me. The shortcut behavior is not evidence of geometric forgetting in my opinion. It is a well-known property of optimization: if a shorter path is available, the optimizer will take it, regardless of what information the deeper layers actually retain. This would happen with any injected signal, not just geometry.
> >
> > That said, I think the paper is okay in showing that there is an issue with information loss (which likely does not apply to geometry only) and that injecting the information helps somewhat. But none of that is surprising, and the framing of "forgetting geometry" is much too strong for the provided evidence. Form that perspective, I do not see the contribution strong enough for ICML level and remain with my score.

---

> > > ### Author Response · Authors · 2026-04-06
> > >
> > > We sincerely thank the reviewer for the continued engagement and for helping us clarify the core message of our work. We also thank the reviewer for agreeing that memory injection 'is reasonable and has effects'. We would like to respectfully address the remaining concerns regarding the interpretation of our findings and the novelty of our contribution.
> > >
> > > ➡️ **1. Geometry is "Differently Encoded" or "Forgotten"**
> > >
> > > We appreciate the reviewer raising the valid alternative hypothesis that geometric information might simply be "differently encoded" in deeper layers rather than lost. However, our empirical results provide strong evidence against this. If the geometry were fully preserved (even if encoded in a highly abstract format), explicitly re-injecting the raw geometric information at deeper layers would be functionally redundant and provide no benefit. Instead, we observe consistent and significant improvements in PDE prediction accuracy across multiple datasets when geometry is re-introduced. This functional improvement indicates that the necessary geometric constraints are genuinely lacking in the deep latent representations, not just hidden.
> > >
> > > Furthermore, this forgetting is not merely a generic information loss. It is a specific consequence of the global mixing operations (FFT and Self-Attention) that define Neural Operators. Unlike locality-preserving CNNs, these global operations progressively "wash out" geometry information.
> > >
> > > ➡️ **2. The Optimization "Shortcut" as Evidence**
> > >
> > > We completely agree with the reviewer that taking a shortcut is a well-known property of optimization. Our argument is precisely that the optimizer's choice to take this shortcut reveals the state of the network. The optimizer will only bypass the physics backbone if the newly injected geometric signal provides a drastically more informative and easier-to-use pathway than the representations currently retained by the deep layers. The fact that the network so aggressively relies on the re-injected geometry highlights just how degraded the native geometric signal had become by that depth.
> > >
> > >
> > > ➡️ **3. Novelty in the Context of Operator Learning**
> > >
> > > We fully agree that mechanisms like skip connections and FiLM are foundational to deep learning. However, our contribution is not the invention of these mechanisms, but rather the identification of a fundamental structural blindspot in the Neural Operator literature. Since the introduction of FNO, the NO community has focused heavily on how to encode complex geometries at the input layer, operating under the assumption that the network will naturally propagate this information.
> > >
> > > Our work is the first to systematically demonstrate that this assumption is flawed. We introduce a diagnostic methodology (mask reconstruction and spectral analysis) to make this "forgetting" measurable, and we apply established techniques (like FiLM) not to claim optimal architecture design, but to prove that counteracting this specific decay resolves the performance bottleneck. While all signals may experience some degradation over depth, geometry is highly measurable to prove this structural flaw.
> > >
> > > We respect that from a general deep learning perspective, the concept of signal decay is familiar. However, identifying how and why this specifically manifests in global operator architectures—and proving that standard sequential skip connections fail to prevent it—provides critical, actionable insights for the PDE and operator learning community. We hope this clarifies the value and specific domain novelty of our contribution.

---

### Official Review · Reviewer_HLG3 · 2026-03-22

**Soundness:** 1
**Presentation:** 2
**Significance:** 2
**Originality:** 3
**Overall Recommendation:** 3
**Confidence:** 3

**Summary:**

The paper analyzes the generalization of neural operators (NOs) across domain geometries and introduces the "geometric forgetting hypothesis." The authors argue that popular NO architectures (such as FNOs and Transolvers) progressively lose geometric information in their deeper layers, attributing this to global mixing mechanisms. To address this, the paper proposes geometric memory injection mechanisms that explicitly condition intermediate layers on encodings of the domain geometry. To validate this hypothesis, the authors also analyze Laplace NOs (LNOs) as a control study, arguing that LNOs possess intrinsic memory.

**Compliance With Llm Reviewing Policy:**

Affirmed.

**Final Justification:**

I increased my score in view of the provided rebuttal. For my remaining concerns, please see my response below.

**Key Questions For Authors:**

- How robust are the results across other architectures (e.g., GINO, GeoFNO, OTNO, Hamlet; see weaknesses above), varying hyperparameters (related to architectures and optimization), and different random seeds?
- Can you provide a more detailed explanation of why the LNO possesses "implicit memory" and discuss the effect of skip connections in the considered NOs?
- Could you further explain why the observations noted in the final "Weaknesses" bullet point above do not contradict the main hypothesis?

**Limitations:**

The paper mentions some limitations. However, several limitations detailed in the "Weaknesses" section above have not been discussed.

**Strengths And Weaknesses:**

**Strengths:**
1. The analysis of geometry-related information propagation in different neural operator architectures is an interesting and relevant problem that has not yet been sufficiently investigated.
2. The authors propose various metrics to validate their "Geometric Forgetting Hypothesis" and provide several ablation studies.

**Weaknesses:**

1. Narrow scope:
    - The FNO does not appear to be the optimal architecture for problems on irregular domains, which is why alternatives like GNO, GeoFNO, OTNO, and GINO have been proposed. Although these works are mentioned in the introduction, they are omitted from the experimental comparisons. Additionally, the use of only a single transformer NO raises the question of whether the current results sufficiently validate the hypothesis.
    - The paper lacks theoretical analysis beyond a basic application of the information-processing inequality. Despite a brief discussion around Equation (4), it remains theoretically unclear why the LNO possesses implicit memory compared to models like the Transolver.
    - The choice of embedding (a concatenation of mask, SDF, and coordinates) appears highly specific. In particular, the mask and SDF seem relevant only to architectures operating on regular grids, which, as noted above, may not be the most suitable tools for the considered problems.
    - The evaluation is relatively small-scale, focusing solely on steady-state problems, utilizing only three architectures, and relying on Cartesian grids.
2. Validity of the results:
    - Although memory injection provides benefits, the improvements might simply result from selecting the best value across noise injections, as standard deviations are not reported for the different experiments. Furthermore, the most effective injection method varies inconsistently.
    - The paper does not appear to discuss the effects of different hyperparameters. Moreover, the role of skip connections (present in both FNO blocks and transformer-based architectures) is completely omitted, despite likely being a key factor in how the model forgets input data, including geometric information.
    - The proposed metrics do not appear as conclusive as the paper suggests: the LNO exhibits a higher MSE loss in later layers than the FNO (regardless of memory injection), the FNO only benefits from memory injection at specific layers, and the Transolver lacks the spectral peaks supposedly "indicative" of boundary information, even after explicit memory injection.

---

> ### Author Rebuttal · Authors · 2026-03-31
>
> We thank the reviewer for the careful and insightful feedback. We are encouraged that, consistent with other reviewers, the **core idea and practical effectiveness of our approach are clearly recognised**, and we address the remaining points below.
>
> ➡️ **Expanded Scope**
>
> Thanks for the suggestion. Here we include **GeoFNO**, which handles irregular domains via coordinate transformation, and **Sp2GNO**, a graph-base dmodel operates directly on irregular meshes. We evaluate on **AirfRANS** (steady-state, irregular mesh) and **CFDBench tube flow** (time-dependent).
>
> | Model | AirfRANS|CFDBench tube|
> |---|---:|---:|
> | GeoFNO (baseline, 4 layer) | 0.0935 | 0.0882 |
> | GeoFNO + memory (0,1) | 0.0835 | 0.0597 |
> | GeoFNO + memory (2,3) | 0.0894 | 0.0672 |
>
> | Model | AirfRANS|
> |---|---:|
> | Sp2GNO (baseline, 6 layer) | 0.2246 |
> | Sp2GNO + memory (0,1,2) | 0.1912 |
> | Sp2GNO + memory (3,4,5) | 0.1948 |
>
> Thus, memory injection is **robust and not limited to a specific configuration**.
>
> ➡️ **Theoretical insufficiency / forgetting analysis**
> We  provide a **theoretically grounded and empirically validated explanation** of a failure mode in neural operators.
>
> Under the Markov structure $G \rightarrow V_1 \rightarrow \cdots \rightarrow V_L$, the Data Processing Inequality establishes that geometry information cannot increase. In Neural Operators, this is compounded by **global mixing (FFT / attention)**, which is not geometry-preserving and progressively weakens boundary-aligned signals.
> We make this effect measurable through layer-wise diagnostics (reconstruction and spectral probes), which show that geometry becomes **less recoverable with depth in practice**. This bridges the gap between theoretical structure (information flow constraint) and observed behaviour.
>
> Crucially, if geometry were fully preserved in latent space, re-injecting it should be redundant or harmful. Instead, we observe consistent improvements—including in geometry-aware models such as GeoFNO, LNO and Sp2GNO—indicating that **implicit propagation is insufficient**.
>
> Thus, our contribution is a **mechanism-level explanation supported by theory and consistent empirical evidence**, together with a simple intervention that corrects it.
>
> ➡️ **Transolver lacks the spectral peaks:** This is expected: Transolver relies on token-wise interactions rather than spectral convolution, so boundary information is not expressed as spectral peaks. Its final layers can still reconstruct sharp boundaries from spectrally smooth features, which is why **mask reconstruction** is the appropriate diagnostic for attention-based models.
>
> ➡️ **LNO exhibits a higher MSE loss than FNO:** While LNO shows higher reconstruction MSE than FNO, it remains **orders of magnitude lower than Transolver** (note log scale in Fig. 3a/5a). Importantly, LNO exhibits **clear spectral peaks**, indicating stronger boundary retention consistent with its Laplace formulation.
>
> ➡️ **FNO only benefits from memory injection at specific layers:** Table 8 demonstrates that for FNO, all injection strategies (Early, Late, All, L0-L3) consistently outperform the baseline model, proving the broad benefit of geometric restoration.
>
> ➡️ **LNO's implicit memory:** We refer LNO's inherent ability to preserve boundary information as 'implicit memory' (peaks in Figure 5b) by its pole and residue formulation. It means that LNO is better designed to remember boundary information. While FNO which use Fourier convolution do not have these spectral peaks (Figure 4a). Therefore, it is reasonable to hypothesis that LNO's superior performance originate from its capability at preserving boundary information.
>
> **➡️ The Role of Skip Connections:**
>
> Standard NO architectures already incorporate ResNet-style sequential skip connections ($v_{t+1}=\sigma(v_t+\mathcal{F}(v_t))$). For instance, the standard FNO update is $v_{t+1}=\sigma(Wv_t+Kv_t)$, where the channel-wise linear transform $W$ acts as a skip connection. Transolver blocks employ similar mechanisms. Because these sequential skip connections are already present in our baselines, they are evidently insufficient to prevent geometric forgetting over many layers.
>
> Our proposed mechanism fundamentally differs from standard skip connections by providing direct memory injection from the **initial** input geometry straight into deep latent representations, bypassing the progressive degradation of layer-by-layer propagation.
>
> ➡️ **Robustness across  hyperparameters, and random seeds:** Memory‑injected variants use same hyperparameters as baseline model without tuning—the fact that we still see consistent gains (15–67% relative error reduction) across four datasets and three architectures demonstrates robustness to hyperparameter choices.
>
> Across three runs, we observe consistent improvements with low variance, indicating that the effect is **stable and not attributable to stochastic training noise**. A note will be included in the supplementary material for completeness.

---

> > ### Author Rebuttal · Reviewer_HLG3 · 2026-04-04
> >
> > Thank you for your response and the additional experiments. I have a couple of follow-up questions:
> >
> > 1. "Memory‑injected variants use same hyperparameters as baseline model without tuning": How are you choosing the hyperparameters of the baseline models? As far as I can see, the original papers did not evaluate their models on all the considered datasets.
> > 2. Why is Sp2GNO not evaluated on the CFDBench tube dataset as well?
> > 3. Thanks for the explanation on the LNO MSE loss. However, I still wonder why LNO is worse than FNO if its intrinsic memory is supposedly so strong? Moreover, could you mathematically explain why the pole-residue formulation "preserves boundary information"?
> > 4. Could you also comment on the highly specific choice of embedding (concatenation of mask, SDF, and coordinates) and the heavy bias of mask and SDF toward regular grids?
> >
> > Thanks in advance!

---

> > > ### Author Response · Authors · 2026-04-04
> > >
> > > We thank the reviewer for the feedback and allowing us to clarify more detail.
> > >
> > > ➡️**1. Hyperparameters choice**
> > >
> > > We agree that since the original baseline papers did not evaluate on all the datasets we considered, we could not simply adopt their default parameters. To ensure a fair and rigorous comparison, for each baseline model on the new datasets, we performed a manual search based on standard heuristics over a reasonable search space using a held-out validation set. To strictly isolate the effect of our architectural contribution, we applied these exact same baseline-optimized hyperparameters to our memory-injected variants without any method-specific tuning. The fact that memory injection yields consistent gains under settings explicitly tuned for the baselines demonstrates its robust, intrinsic architectural benefit without relying on separate hyperparameter optimization.
> > >
> > > ➡️ **2. Sp2GNO on CFDBench tube flow dataset**
> > >
> > > Sp2GNO, as a graph-based operator, is computationally intensive  (>12h per training run). Within the rebuttal timeframe, we prioritised experiments that most directly address the reviewer’s concern regarding generalisation. By evaluating GeoFNO on both AirfRANS and CFDBench, and Sp2GNO on AirfRANS, we successfully demonstrated that our memory injection mechanism generalizes to: (1) geometry aware neural operators of both coordinate transformation type (GeoFNO) and graph type (Sp2GNO), (2) irregular mesh input (AirfRANS), and (3) time dependent problems (CFDBench tube flow). We believe these additions comprehensively validate our geometric forgetting hypothesis across diverse architectures and data regimes.
> > >
> > > ➡️ **3.LNO's intrinsic memory**
> > >
> > > We thank the reviewer for this perceptive question, which allows us to clarify a key distinction in how geometric memory manifests in different architectures.
> > >
> > > To comprehensively evaluate geometric forgetting, our paper deliberately introduces two distinct, complementary diagnostic metrics: (1) Macroscopic Geometry Retention, measured by the spatial mask reconstruction MSE (Definition 3.3), (2) Boundary Information Retention for spectral based method, measured by the preservation of high-frequency spectral peaks (Definition 3.4).
> > >
> > > It is true that standard LNO exhibits a slightly higher Mask MSE than standard FNO in deeper layers. However, this must be viewed in context. First, LNO’s reconstruction MSE remains well within a stable, functional regime (compared with Transolver). Second, and most importantly, LNO’s "strong intrinsic memory" primarily manifests in the spectral domain. As shown in our spectral analysis (Figure 5b), standard LNO naturally preserves the sharp spectral peaks that correspond to boundary information. FNO, by contrast, completely washes out these boundary signals in its deeper layers unless external memory is injected (Figure 4a). We do not claim LNO is "perfect" at spatial mask reconstruction, but rather that its inherent ability to preserve these critical spectral boundary peaks without any external injection demonstrates a highly effective, intrinsic form of geometric memory.
> > >
> > > To further clarify 'why the pole-residue formulation preserves boundary information', we follow the original LNO paper.
> > > For a signal $v(t)=\sum_{l=-\infty}^{\infty}\alpha_l\exp(i\omega_lt)$, after Laplace convolution with Laplace domain kernel $K(s)=\sum_{n=1}^N\frac{\beta_n}{s-\mu_n}$, it becomes $u_L(t)=\sum_{n=1}^N\gamma_n\exp(\mu_nt)+\sum_{l=-\infty}^{\infty}\lambda_l\exp(i\omega_lt)$. The first summation in Laplace convoluted $u_L(t)$ represent transient response (exponential decay), which is absent in Fourier convolution and translate to boundary effect in spatial domain (replace $t$ with $x$).
> > >
> > > ➡️**4. The choice of input**
> > >
> > > We thank the reviewer for the opportunity to clarify this. We respectfully note that the choice of geometric embedding (coordinates, SDF, and mask) is highly standard in the context of computational physics and is fundamentally grid-agnostic.
> > >
> > > First, in standard PDE workflows generated by traditional numerical solvers (e.g., COMSOL, OpenFOAM), these geometric descriptors are readily available or easily computable for any arbitrary set of control nodes during mesh generation or simulation setup. These form a highly informative descriptor of the problem concerned and could facilitate operator learning. Because they are evaluated pointwise at specific node locations, they possess no inherent structural bias toward regular Cartesian grids (which we utilized initially only to ensure fair comparison with standard FNO/Transolver baselines).
> > >
> > > Our new experiments directly validate this compatibility with irregular geometries. The AirfRANS dataset utilizes highly irregular meshes where these exact pointwise features are defined on scattered nodes. As shown, memory injection successfully leverages these embeddings to improve both GeoFNO and Sp2GNO performance on purely irregular meshes.

---

### Official Review · Reviewer_ndjS · 2026-03-24

**Soundness:** 3
**Presentation:** 4
**Significance:** 2
**Originality:** 2
**Overall Recommendation:** 5
**Confidence:** 3

**Summary:**

Deep neural operators (NOs) are a promising route to cost-effective simulation of physical systems, and have attracted a significant literature in recent years. By learning neural operators from infinite-dimensional spaces of conditioning functions to spaces of solutions, the field hopes to find neural networks which encode relevant features of domain geometry and PDE behaviour.

One of the most important inputs to these NOs is an encoding of the domain geometry. The paper under review identifies a key problem in two of the existing techniques (FNO, Transolver), namely, that information about the domain geometry is attenuated throughout the network and, as a result, the learned solutions do not treat the boundary conditions correctly. They experiment with some straightforward means of re-injecting this information throughout the network, and do a solid job of arguing that this simple architectural change significantly improves the performance of FNO and Transolver. In contract, LNO does not see an improvement: they link this to the nature of the Laplace transform underlying the technique.

In brief, I would summarise the paper as follows: FiLM conditioning works well for neural operators.

**Compliance With Llm Reviewing Policy:**

Affirmed.

**Final Justification:**

The rebuttal responded to at least one of my primary concerns. I was not convinced by the response on the theoretical points, but the additional experiment was enough for me to increase my score.

**Key Questions For Authors:**

Questions/comments:

* It seems plausible that if a single network is trained on multiple different domain geometries, encoded in some way, then it may internally learn some kind of similar universal solution as Geo-FNO does (this would be analogous to the results showing that language models translate many languages into a joint internal language). In this case re-injecting the original domain geometry (which has not been “standardised” in the forward pass) is unlikely to be useful or even harmful. The results in paper seem to provide partial evidence *against* such a universal representation of domain geometry in FNO and Transolver.
* The best method is FiLM which introduces a multiplicative gating into e.g. the Transformer. This seems to potentially undermine some of the benefits of using a transformer architecture in the first place, can the authors comment on whether FiLM impacts the scalability of the architecture (it is my impression this is one of the key reasons for introducing neural operators in the first place).

**Limitations:**

yes

**Strengths And Weaknesses:**

Strengths:

The paper is well-written and argued, and makes an incremental but potentially useful improvement on an important fundamental primitive in AI for Science. It considers a range of models (FNO, Transolver, LNO) and a range of ways of injecting the geometric information, and carefully examines the premise for why its architectural innovation could be expected to improve performance. Of course, the main strength is that percentage gains for FNO and Transolver are significant.

Weaknesses:

* The first named contribution, to introduce "geometric forgetting" and characterize it theoretically, does not seem substantial to me. This is a generic property of information processing by neural networks (as their citing of the data processing inequality shows).
* The main strength of the paper is therefore that the architectural intervention works, but the effect is only pronounced on FNO and Transolver. Indeed, consulting Table 2 we see that for many of the tasks FNO+memory and Transolver+memory are roughly competitive with LNO as a baseline. Since this method is, in the author's own description, already less sensitive to forgetting the domain geometry, it raises the question of why we need this memory injection: just use LNO. Having said that, there may be reasons to use FNO or Transolver that I am unaware of, and in those situations this architectural remedy may be welcomed by the community.
* If I understand correctly Geo-Fno (cited as Li et al) aims to solve this irregular domain problem by learning a transformation to and from a standard domain geometry. Why is there no comparison to Geo-FNO?

---

> ### Author Rebuttal · Authors · 2026-03-31
>
> We thank the reviewer for the thoughtful and constructive feedback. **We appreciate that the reviewer, consistent with others, recognises the practical value of our contribution**, and we address the remaining points below.
>
> ➡️ **“Geometric forgetting” is generic (DPI)**
> Geometric forgetting is beyond a generic DPI effect. It arises specifically from **global mixing (FFT / attention)** in Neural Operators, which is not geometry-preserving and progressively weakens boundary-aligned signals—unlike locality-preserving CNNs. Prior work addresses geometry **encoding at input**, not its **persistence through depth**. We explicitly diagnose this via mask reconstruction and spectral analysis, and show that restoring geometry access consistently improves PDE accuracy.
>
>
>
> ➡️ **“Why not just use LNO?”**
> Thanks-- This addresses a different question. LNO is an architectural choice, whereas our contribution targets **how geometry is preserved through depth**, which is orthogonal.
>
> While LNO better captures boundary effects (through pole and resudue formulation), it still relies on implicit propagation and does not provide **explicit access to geometry at deeper layers**. If this were sufficient, additional conditioning should not help—yet we observe consistent gains even for LNO **(main Table 2, and Appendix B.2 Table 8)**.
> This indicate the effect is **structural, not architecture-specific**. Our method is therefore complementary and directly applicable to widely used operators such as FNO and Transformer-based models.
>
>
>
>
> ➡️ **No comparison to GeoFNO / universal geometry representation**
> We have added GeoFNO experiments (AirfRANS, steady-state), which maps inputs to a regular grid:
>
> | Model | Relative L2 |
> |---|---:|
> | GeoFNO (baseline) | 0.0935 |
> | GeoFNO + memory (0,1) | 0.0835 |
> | GeoFNO + memory (2,3) | 0.0894 |
>
>
> This indicates that the effect is **robust and not tied to a specific configuration**.
>
> Despite explicit geometry normalisation, memory injection still improves performance. This suggests that geometry is not fully preserved in latent representations, and that **persistent access to geometry throughout depth is beneficial in general**.
>
> This directly addresses the “universal geometry representation” hypothesis. If geometry were fully internalised in latent space (as suggested), re-injecting the original geometry should be redundant or harmful. Instead, we observe consistent improvements, indicating that **geometry is not fully preserved through depth in practice**, even in geometry-aware operators.
>
> This supports our claim that **explicit access to geometry remains beneficial beyond input encoding**.
>
>
>
>
>
> ➡️ **Does FiLM affect scalability?**
> No. FiLM adds **channel-wise affine modulation**, i.e. $O(C)$ per token, which is negligible compared to the dominant costs of Neural Operators: $O(N \log N)$ for FFT-based layers and $O(N^2)$ (or $O(N)$ with efficient variants) for attention.
>
> Importantly, FiLM does **not introduce additional token interactions or alter the mixing operator**—it only modulates features locally. Therefore, it does not affect the scaling behaviour that motivates Neural Operators.
>
> Moreover, the improvement is not specific to FiLM: we observe similar gains with additive and concatenation-based injection. This shows that the benefit comes from **persistent access to geometry**, not from multiplicative gating itself, and can be implemented without impacting scalability.

---

### Decision · Program_Chairs · 2026-04-30

**Decision:**

Accept (regular)

**Comment:**

This paper investigates the progressive loss of geometric information in deep neural operators (NOs), attributing this degradation to global mixing mechanisms like FFT and self-attention. To counteract this "Geometric Forgetting Hypothesis," the authors propose a lightweight geometry memory injection mechanism and diagnose the information loss through layer-wise geometric probing. Initially, reviewers commended the paper's motivation, the simplicity of the proposed solution, and the clever use of the Laplace Neural Operator (LNO) as a control study. However, multiple reviewers raised valid concerns regarding the initially narrow empirical scope (e.g., lack of comparison with geometry-aware baselines like GeoFNO, limitation to steady-state problems on regular grids) and questioned whether the phenomenon was specifically "geometry forgetting" or just a generic manifestation of information decay over deep layers.

The authors provided a strong rebuttal that broadened the experimental scope. They successfully demonstrated the robustness of their memory injection method on irregular meshes (AirfRANS), time-dependent problems (CFDBench), and additional geometry-aware architectures (GeoFNO and Sp2GNO). These additions resolved the primary concerns of Reviewers ndjS, HLG3, and QeWr, who recognized the practical value and consistent empirical gains (15-67% error reductions). However, Reviewer nJuC remained explicitly unconvinced by the theoretical framing and maintained a Reject score. While nJuC conceded that the empirical results were "solid and potentially useful," they firmly argued that the paper misinterprets generic information loss (governed by the Data Processing Inequality) as a geometry-specific phenomenon, noting that the theoretical justification claims more than the evidence strictly supports.

Despite the valid philosophical disagreement raised by Reviewer nJuC regarding the terminology and theoretical framing, the core technical contribution of the paper remains robust. The authors have demonstrated a structural bottleneck in widely used global operator architectures and provided an effective, computationally inexpensive remedy. The supplementary experiments successfully demonstrated that the method generalizes across diverse data regimes and architectures. Because the primary remaining objection centers on narrative framing rather than technical or empirical flaws, and because the practical utility to the AI-for-science community is exceptionally clear, I recommend an Accept for this submission.